# Scale-free behavioral dynamics directly linked with scale-free cortical dynamics

**Sabrina A Jones, Jacob H Barfield, V Kindler Norman, Woodrow L Shew***

Department of Physics, University of Arkansas at Fayetteville, Fayetteville, United States

**Abstract** Naturally occurring body movements and collective neural activity both exhibit complex dynamics, often with scale-free, fractal spatiotemporal structure. Scale-free dynamics of both brain and behavior are important because each is associated with functional benefits to the organism. Despite their similarities, scale-free brain activity and scale-free behavior have been studied separately, without a unified explanation. Here, we show that scale-free dynamics of mouse behavior and neurons in the visual cortex are strongly related. Surprisingly, the scale-free neural activity is limited to specific subsets of neurons, and these scale-free subsets exhibit stochastic winner-take-all competition with other neural subsets. This observation is inconsistent with prevailing theories of scale-free dynamics in neural systems, which stem from the criticality hypothesis. We develop a computational model which incorporates known cell-type-specific circuit structure, explaining our findings with a new type of critical dynamics. Our results establish neural underpinnings of scale-free behavior and clear behavioral relevance of scale-free neural activity.

## Editor's evaluation

This paper is an important study that is of interest to neuroscientists studying the organization of neural activity and behavior. The authors present compelling evidence to link the apparently scale-free distributions of behavioral metrics with scale-free distributions of neural activity. They then explore computationally mechanistic models that could account for these observations. The simulations of mechanistic models are provocative and suggest interesting network-connectivity hypotheses to test in future experiments.

*For correspondence:
shew@uark.edu

## Introduction

From fidgeting to expeditions, natural body movements manifest over a very broad range of spatiotemporal scales. The complexity of such movements is often organized with fractal structure, scale-free fluctuations spanning multiple spatiotemporal orders of magnitude (*Anteneodo and Chialvo, 2009*; *Proekt et al., 2012*; *Sims et al., 2008*; *Viswanathan et al., 1999*). Such behavioral complexity may be beneficial for foraging (*Garg and Kello, 2021*; *Sims et al., 2008*; *Viswanathan et al., 1999*; *Wosniack et al., 2017*), visual search (*Viswanathan et al., 1999*), decision-making based on priority (*Barabási, 2005*; *Sorribes et al., 2011*), flexible switching of behavior (*Abe, 2020*), and perhaps more. Similarly, fluctuations of ongoing neural activity in the cerebral cortex can exhibit fractal, scale-free fluctuations like the spatiotemporal cascades sometimes referred to as 'neuronal avalanches' (*Beggs and Plenz, 2003*; *Bellay et al., 2015*; *Ma et al., 2019*; *Priesemann et al., 2014*; *Scott et al., 2014*; *Shew et al., 2015*; *Shriki et al., 2013*; *Tagliazucchi et al., 2012*; *Yu et al., 2017*) and long-range temporal correlations (*Hardstone et al., 2012*; *Kello et al., 2010*; *Palva et al., 2013*; *Smit et al., 2013*). Considering these observations of behavior and brain

**eLife digest** As we go about our days, how often do we fidget, compared to how frequently we make larger movements, like walking down the hall? And how rare is a trek across town compared to that same walk down the hall? Animals tend to follow a mathematical law that relates the size of our movements to how often we do them.

This law posits that small-to-medium movements and large-to-huge movements are related in the same way, that is, the law is 'scale-free', it holds the same for different scales of movement. Surprisingly, measurements of brain activity also follow this scale-free law: the level of activation of a group of neurons relates to how often they are activated in the same way for different levels of activation.

Although body movements and brain activity behave in a mathematically similar way, these two facts had not previously been linked. Jones et al. studied body movements and brain activity in mice, and found that scale-free body movements were linked to scale-free brain activity, but only in certain subsets of neurons. This observation had been hidden because other subsets of neurons compete with scale-free neurons. When the scale-free neurons turn on, the competing groups turn off. When averaged together, these fluctuations cancel out.

The findings of Jones et al. provide a new understanding of how brain and body dynamics are orchestrated in healthy organisms. In particular, their results suggest that the complex, multi-scale nature of behavior and body movements may emerge from brain activity operating at a critical tipping point between order and disorder, at the edge of chaos.

activity together, a simple, yet unanswered question arises. Are scale-free dynamics in cerebral cortex related to scale-free body movements? Or are these statistical similarities merely superficial, without any direct link?

Two previous studies in humans have shown that temporally scale-free ongoing brain dynamics are related to some subtle aspects of behavior – fluctuations in success at rhythmic finger tapping (*Smit et al., 2013*) and sensory detection tasks (*Palva et al., 2013*). But, here, we are concerned with less subtle, larger-scale body movements and invasive measurements of brain activity with single neuron resolution.

According to prevailing theories of scale-free ongoing neural activity, these dynamics are often interpreted as 'background' activity, not directly linked to behavior. This view arises in part because computational models of scale-free neural activity generate this type of dynamics autonomously, due to internal interactions, without explicit modeling of a body that behaves (*Dalla Porta and Copelli, 2019*; *di Santo et al., 2018*; *Girardi-Schappo et al., 2020*; *Li and Shew, 2020*; *Wilting et al., 2018*). Similarly, in experiments, scale-free neural activity has typically been observed in animals that are at rest, not engaged by any particular motor task or sensory input, or under anesthesia (*Fontenele et al., 2019*; *Gautam et al., 2015*; *Gireesh and Plenz, 2008*), or in vitro (*Beggs and Plenz, 2003*; *Bellay et al., 2020*; *Shew et al., 2009*; *Tetzlaff et al., 2010*) where there is no behavior to be observed. Based on these previous experimental observations and modeling efforts, one might naturally conclude that scale-free neural activity is internally generated, emerging spontaneously, without any link to behavior.

However, recent experiments in awake mice have revealed that behaviors typically ignored by experimenters, such as whisking, eye and face movement, fidgeting, walking, and running, are strongly related to ongoing neural activity in the cortex (*Clancy et al., 2019*; *Musall et al., 2019*; *Salkoff et al., 2020*; *Stringer et al., 2019*). These studies did not address our question here about scale-free-ness of neural activity and behavior, but they suggest that if ongoing activity is scale-free, then it may be strongly related to behavior. More specifically, these studies point out the possibility that previous observations of scale-free neural activity in awake animals didn't notice a link to behavior because those studies did not measure ongoing, spontaneous movements of the face, whiskers, and body. If this is the case, it would dramatically revise the typical interpretation of scale-free ongoing activity and provide direct evidence that scale-free behavior and scale-free brain activity are inseparably related. However, in the work by *Stringer et al., 2019*, the measured behaviors could explain only 10–30% of the total variance in neural activity fluctuations. Thus, the question remains: is this behaviorally-related 10–30% of neural activity scale-free? Here, our aim was to answer this question.

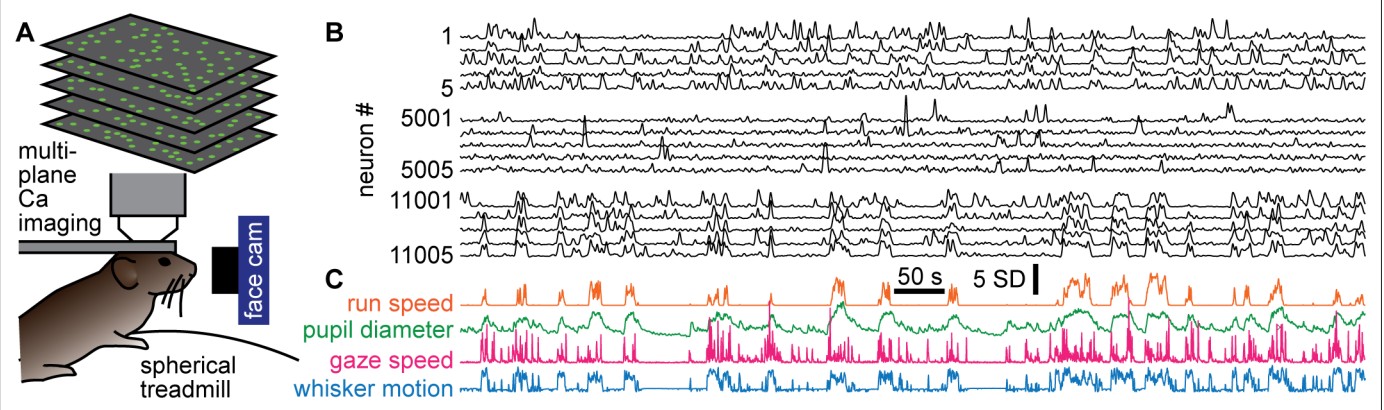

**Figure 1.** Simultaneous recording of behavior and neural activity. (**A**) Multiplane calcium imaging measured ~10,000 neurons in shallow layers of mouse visual cortex, while a camera measured facial motion and a treadmill measured run speed. (**B**) Example activity time series from 15 neurons (out of 11,005 total in this recording). Notice some neurons were strongly correlated with behavior (bottom five), while others were anticorrelated (top five), and others were nearly uncorrelated (middle five). (**C**) The four aspects of behavior - run speed, pupil diameter, speed of changes in gaze direction, and whisker motion - tended to covary on long time scales, but differ in fast fluctuations.

## Results

Here, we analyzed simultaneous recordings of more than 10,000 pyramidal neurons during 1–2 hr in visual cortex of awake mice ( seven mice, nine recordings, recording duration 1.52 ± 0.51 hr), performed and first reported by *Stringer et al., 2019*. The neurons were distributed throughout a volume (300–400 μm in-depth, 0.9 × 0.935 mm in the area) and sampled at 3 Hz. In addition to the neural activity, several aspects of behavior and body movements were also recorded (*Figure 1*). Here, we focused on four aspects of behavior. First, we studied run speed, which was assessed using an optical sensor and a floating spherical treadmill. Second, we examined pupil diameter, which was obtained from a camera targeting the faces of the mice. Pupil diameter dynamics are associated with changes in arousal and other aspects of body movement and sensory input (*Reimer et al., 2016*; *Reimer et al., 2014*; *Vinck et al., 2015*). Third, we studied changes in direction of gaze by tracking the speed of the center of the pupil as the mice looked around. Finally, we used previously developed methods to study whisker motion (*Stringer et al., 2019*), which was also obtained from the face camera.

### Scale-free behavior

The behavior tended to occur in bursts; rest periods were punctuated with well-defined bouts of body movement (*Figure 1C* and *Figure 2A*). All four of the behavioral variables we studied tended to start and stop together, approximately, but differed in their detailed, fast fluctuations (*Figure 1C*). We first sought to determine whether the behaviors exhibited scale-free dynamics. To this end, we first defined each bout of elevated body activity, hereafter called a behavioral 'event,' based on a threshold (the median of the entire time series). Each behavioral event began when the behavioral time series, run speed for example (*Figure 2A*), exceeded the threshold and ended when it returned below the threshold. The 'size' of each behavioral event was defined as the area between the threshold and the variable during the event (*Figure 2A*). This definition of events and their sizes is motivated by previous studies of scale-free neural activity (*Gautam et al., 2015*; *Larremore et al., 2014*; *Li and Shew, 2020*; *Poil et al., 2012*), typically referred to as neuronal avalanches. In the case of run speed, the event size corresponds to an effective distance traveled during the event. If the animal behavior is scale-free, one expects the distribution Pr(s) of behavioral event sizes (s) to have a power-law form, Pr(s)~$s^{-\tau}$. We found that these distributions often were in the form of a power-law over a wide range of event sizes (*Figure 2B*). For example, the run speed event size distribution for mouse 1 was scale-free for nearly four orders of magnitude of event sizes (power-law range, r=3.9 decades). To rigorously assess the range over which the data is power-law distributed, we used a maximum likelihood fitting algorithm that accounts for the number of events observed and the possibility of a cutoff at the head and tail of the distribution (Methods). Our approach builds on that used in previous studies (*Shew et al., 2015*).

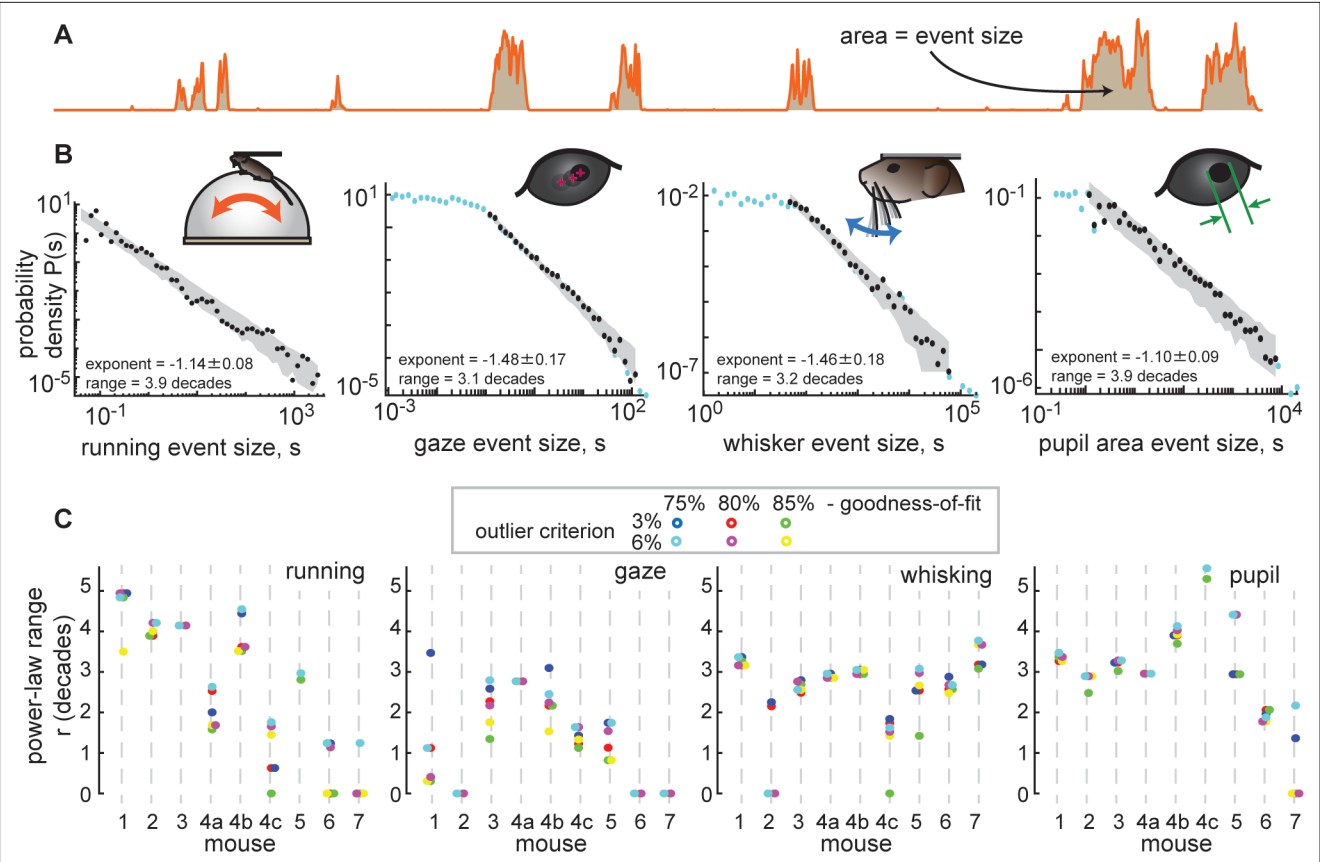

**Figure 2.** Spontaneous behavior is often scale-free. (**A**) Behavioral events (run speed, in this example) are defined by excursions above the median; event size is the area (shaded patch) between the curve and the median. (**B**) Four examples of behavioral event size distributions with large power-law range $r>3$ dB. Black points indicate the part of the distribution that is fit well by a power law. Blue points fall outside the power-law range. Gray patch indicates the expected variability (10th to 90th percentile) of the best-fit power law. (**C**) Summary of power-law range for all mice and all behaviors. Gaze dynamics were least likely to have a large power-law range. All mice had at least one behavior with a very large (>3 decades) power law range. Changes in power-law fitting parameters (goodness-of-fit criteria and outlier criteria) did not change our general conclusions.

The online version of this article includes the following figure supplement(s) for figure 2:

**Figure supplement 1.** Interpreting probability density functions (PDF) with logarithmic bins.

**Figure supplement 2.** Outlier exclusion effects on power-law fitting algorithm.

We report the power-law range r in decades, i.e., the number of orders of magnitude. If no range meets our fit criteria for statistical significance, we report r=0.

Although not all behaviors were scale-free over such a large range for all recordings, we found that most of the mice exhibited scale-free behaviors, particularly for running and pupil fluctuations (*Figure 2C*). In addition to the power-law range for larger behavioral events, some behaviors exhibited a range of small-scale behavioral events with a flat distribution consistent with low amplitude noise in the measurement methods (*Figure 2—figure supplement 1*). These conclusions were robust to changes in the two most important parameters for the power-law fitting algorithm: the goodness-of-fit criterion and the outlier exclusion threshold (more on outlier exclusion in *Figure 2—figure supplement 2*).

## Scale-free neural activity

Next, we turned to the neural activity to test whether it was scale-free like the behavior. First, we averaged over all neurons to obtain a single population activity time series. Then, we treated the neural data in a similar way to the behavioral data. We defined neural events with a median threshold and defined event size as the area between the threshold and the data during the event (*Figure 3A*). This definition of events and event sizes is well-established in previous studies of scale-free neural activity

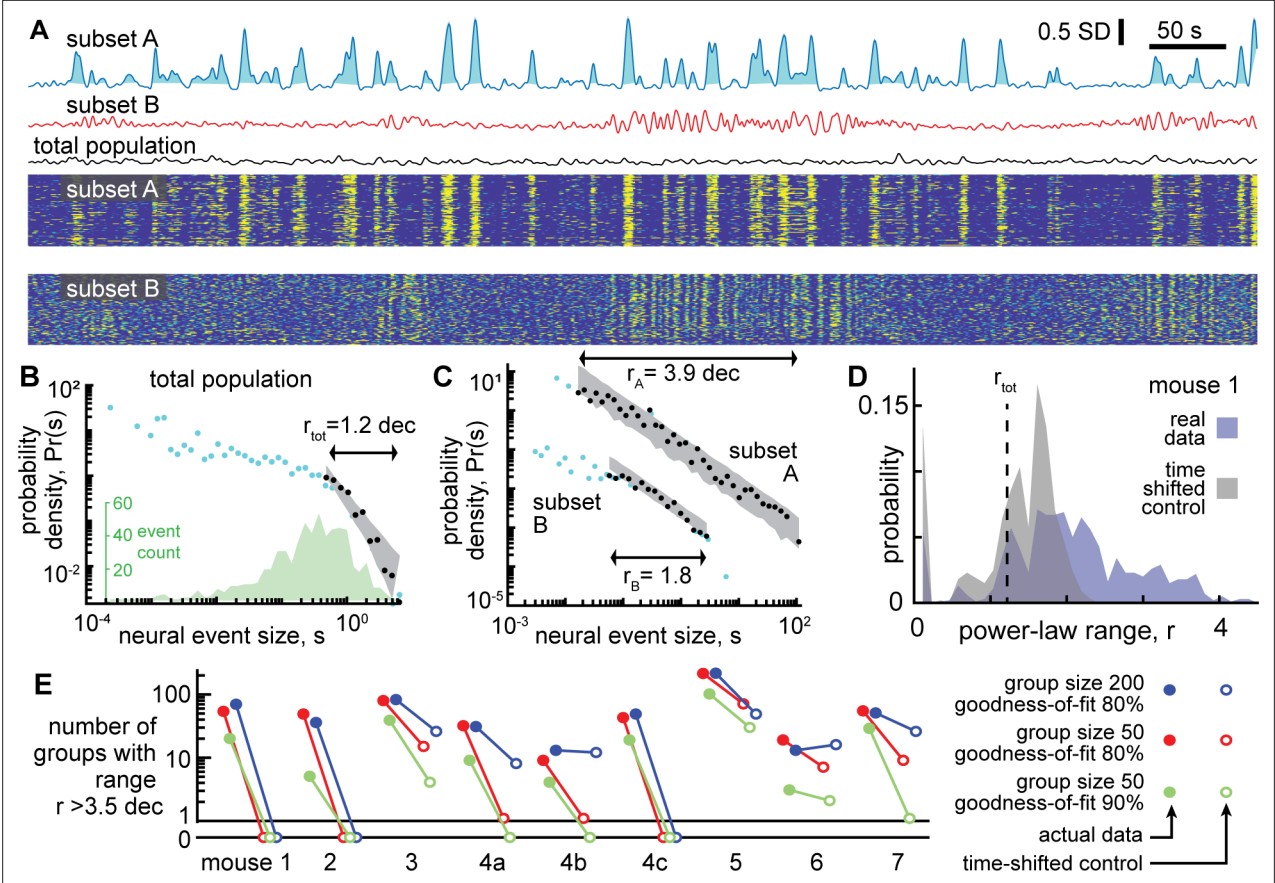

**Figure 3.** Certain subsets of neurons exhibit scale-free dynamics, but total population does not. (**A**) Activity of the total neural population (black) and two subsets of neurons (blue, red) are shown. Each excursion above the median is one neural event (blue shaded area for subset A). Examples in panels A, B, and C are from mouse 1. (**B**) For the total population, the distribution of neural event sizes is not well fit by a power law (small range $r_{tot}$ = 1.2 decades). Histogram of event sizes (green, same bins used for blue, black distributions) shows that head of the distribution is noisy because very few events had very small sizes. (**C**) Certain subsets of neurons exhibited very large power-law range ($r_A$ = 3.9 decades), while other subsets did not ($r_B$ = 1.8 decades). These two example distributions are offset vertically to aid comparison of shape. (**D**) Distribution of power-law range $r$ for 1000 subsets from mouse 1 (blue). Power law range $r$ reached 4.5 decades for some subsets but did not exceed 2.6 decades for time-shifted controls (gray) in this mouse. (**E**) Summary of all mice, showing the number of subsets (out of 1000) with power-law range $r$>3.5 decades, compared to time-shifted controls. Results were qualitatively unchanged for different group size or goodness-of-fit criteria.

The online version of this article includes the following figure supplement(s) for figure 3:

**Figure supplement 1.** Neural event size distributions are robust to some loss of spikes due to analysis of Ca imaging signals.

**Figure supplement 2.** Reconciling fast and slow time scales of electrophysiology and Ca imaging.

(*Gautam et al., 2015*; *Larremore et al., 2014*; *Li and Shew, 2020*; *Poil et al., 2012*). (For more discussion of why this definition was chosen over other established alternatives, see Methods). The resulting distributions of neural event sizes were not scale-free; they were poorly fit by a power-law distribution (*Figure 3B*). Considering all 9 recordings, we found that r=1.5 ± 0.9 decades (mean ± SD). Such a short power-law range should be interpreted as evidence against any power-law at all, because nearly any function can be fit on a short range of data (more on why a large power-law range is important below).

One note of warning: at first glance, the distribution in *Figure 3B* may appear to have a decent power-law fit for the small event sizes. This is a misleading artifact resulting from using logarithmic bins and very few samples to estimate probability density. The inset in *Figure 3B* shows the number of events rather than probability density. See also *Figure 2—figure supplement 1* for more clarification of this point. We emphasize that our power-law fitting algorithm does not use any bins and accounts for sample size, thus avoiding such misleading artifacts.

The lack of scale-free dynamics for the neural population was surprising considering two facts together: first, the behavior exhibited scale-free dynamics (*Figure 2*), and second, many neurons are strongly correlated with behavior (*Figure 1* and *Stringer et al., 2019*). One possible explanation for this puzzling observation is that summing the entire population together obscures the scale-free dynamics of certain subsets of neurons. Next, we set out to test this possibility. Since we did not know, a priori, which neurons might be in these subsets, we adopted a brute-force, shotgun search. First, we picked a 'seed' neuron at random from the entire population. Then we identified the 50 neurons that were most correlated with the seed neuron (we also tried 200 neurons without major impacts on the following results, *Figure 3E*). We averaged the activity of these 50 neurons to obtain a single time series and proceeded to define events and examine their size distribution. Two examples of time series obtained from subsets of neurons in this way are shown in *Figure 3A*. We repeated this process for 1000 different seed neurons. We found that some subsets of neurons were indeed scale-free, with power-law scaling over more than four orders of magnitude, while other subsets were not scale-free (*Figure 3C*). With so many neurons to choose from and a shotgun approach like this, it is important to avoid chance-level false positive conclusions. As a conservative control for this possibility, we repeated the analysis, but with surrogate data, generated by applying a random time shift to each neuron's activity time series relative to other neurons. This time-shifted control data, thus, has identical statistics to the original data at the single neuron level, but correlations among neurons occur only by chance. For time-shifted controls, we found that large power-law ranges (greater than 3.5 decades) were rare and not found at all in the majority of recordings (*Figure 3D and E*). This conclusion was robust for different choices of group size and goodness-of-fit criteria used for the power-law fitting (*Figure 3E*).

Two concerns with the data analyzed here are that it was sampled at a slow time scale (3 Hz frame rate) and that the deconvolution methods used to obtain the data here from the raw GCAMP6s Ca imaging signals are likely to miss some of the underlying spike activity (*Huang et al., 2021*). Since our analysis of neural events hinges on summing up activity across neurons, could it be that the missed activity creates systematic biases in our observed event size statistics? To address this question, we analyzed some time-resolved spike data (Neuropixel recording from *Stringer et al., 2019*). Starting from the spike data, we created a slow signal, similar to that we analyzed here by convolving with a Ca-transient, down sampling, deconvolving, and z-scoring (*Figure 3—figure supplements 1–2*). We compared neural event size distributions to 'ground truth' based on the original spike data (with no loss of spikes) and found that the neural event size distributions were very similar, with the same exponent and the same power-law range (*Figure 3—figure supplement 1*). Thus, we conclude that our reported neural event size distributions are reliable.

## Linking scale-free brain activity and behavior

Having established that certain subsets of neurons have scale-free dynamics and that behavior has scale-free dynamics, we next sought to assess how the neural and behavioral dynamics are related. We first computed the correlation coefficient between each behavior time series and each neural subset time series. The correlation was very weak between the total neural population time series and behavior (red x's in *Figure 4*). For the 50-neuron subsets, we found a very wide range of behavioral correlation values, from 0.9 to –0.75 (*Figure 4*). Interestingly, the subsets with the greatest correlation with behavior also tended to have the largest ranges of power-law scaling. Moreover, the neural subsets with the smallest power-law range were most often near zero correlation with behavior. These results strongly suggest that the scale-free subsets of neurons are directly related to scale-free behavior. It is also interesting to note that many of the subsets that were strongly anticorrelated with behavior also had a large power-law range (the U-shaped relationships in *Figure 4*). These findings were very robust in 6 out of 9 recordings (mouse 1–4 c, *Figure 4*), but less prominent for some behaviors in the remaining three recordings (mouse 5–7). We note that these three 'outlier' animals also had tendencies to run much more than usual (mouse 6 and 7 ran 50 times more than the next most active animal) or much less than usual (mouse 5 ran 10 times less than the next least active animal) as shown in *Figure 4—figure supplement 1*. This observation suggests that our main findings hold best in vigilance states that are moderate, not too hyperactive, or hypoactive.

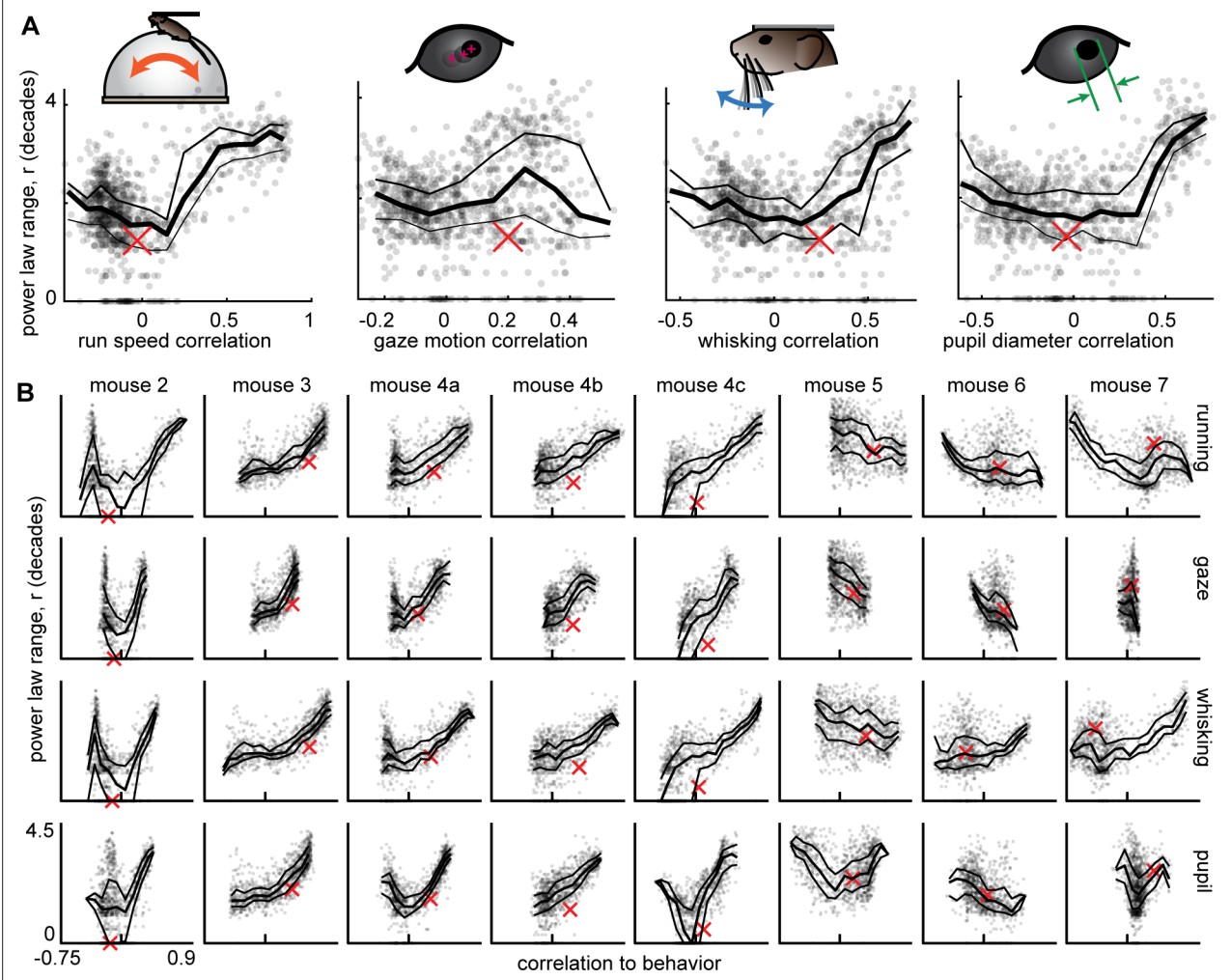

**Figure 4.** Scale-free neural subsets correlate with behavior, but non-scale-free subsets do not. (**A**) Each point represents the power-law range r and behavioral correlation of one neural subset from mouse 1. Subsets with large r tend to be most correlated with behavior. Thick line is a moving average of points; thin lines delineate quartiles. The red x represents the total population. (**B**) Same as panel A for the other eight recordings. Tick mark on horizontal axis marks zero correlation. Points have partially transparent markers, thus darker areas reveal higher density of points.

The online version of this article includes the following figure supplement(s) for figure 4:

**Figure supplement 1.** Outlier recordings.

## Power-law range, exponents, and scaling relations

Previous studies of scale-free neural activity have typically not reported power-law range. Readers may wonder why we focused our analyses here on the power-law range. So far, our results offer two answers to this question. First, and most importantly, only neural subsets with large power-law ranges were strongly correlated with behavior (*Figure 4*). Second, any claim of power-law scaling is stronger if that scaling extends over a greater range. Previous studies often rely on a 'rule of thumb' that one or two decades of scaling is not bad. For the data analyzed here, we found that a power-law range of 1–2 decades is insignificant compared to chance for event size distributions (*Figure 3*).

One reason previous studies of scale-free brain activity have not reported power-law range stems from their motivating hypothesis that scale-free activity results from the neural circuits operating in a special dynamical regime near the tipping point of a phase transition, i.e., near criticality. Rather than power-law range, these studies focused on scaling relationships between the power-law exponents of the event size and duration distributions as predicted by the theory of critical phenomena. Based on our results, we contend that these previous studies might be strengthened by also accounting for power-law range. Conversely, previous studies also raise a natural question about whether the data

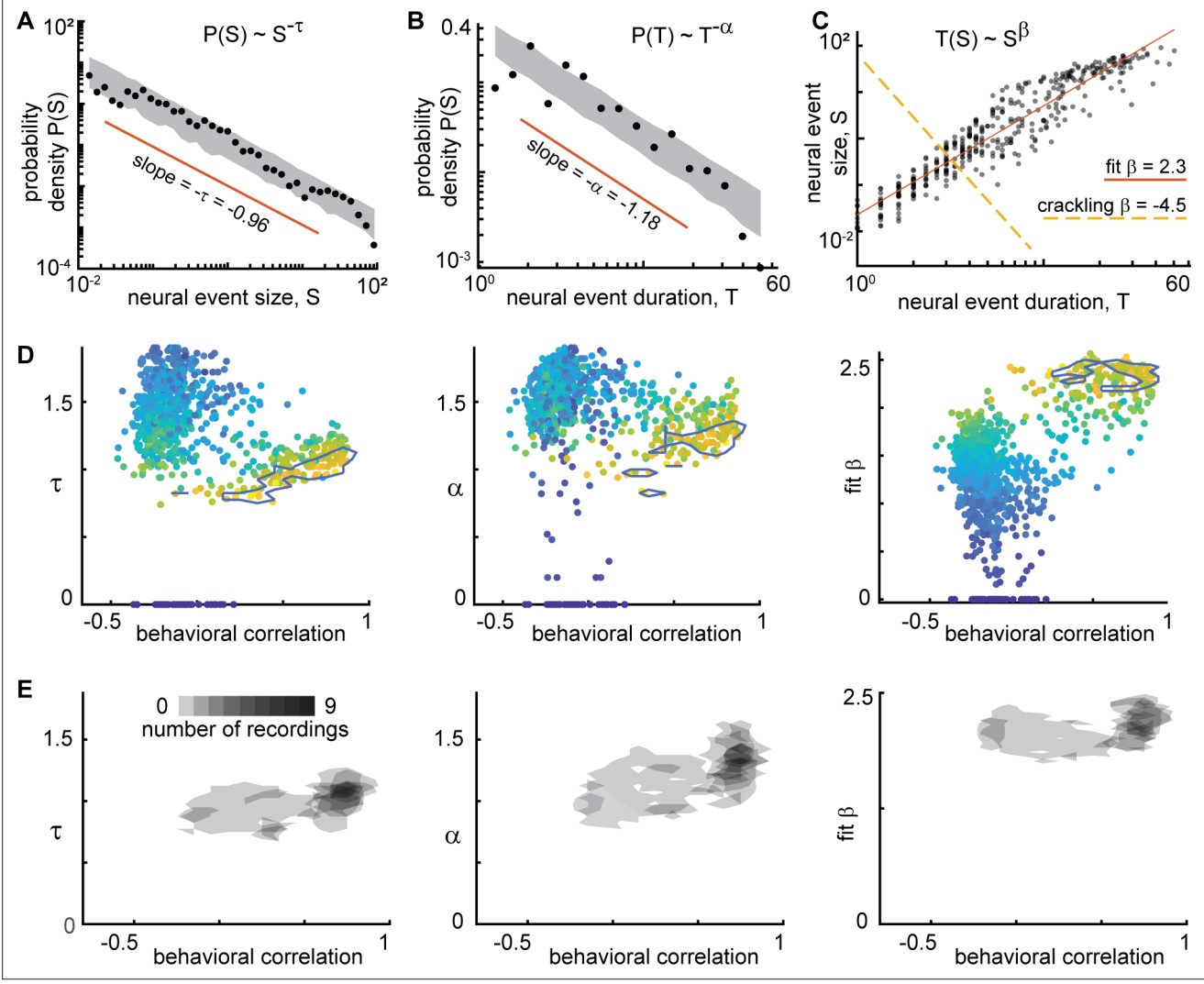

**Figure 5.** Consistent scaling relations for behaviorally-correlated neural subsets. (**A**) Example neural event size distribution with exponent $\tau$=0.96 of the best fit power law (slope on log-log plot). (**B**) Distribution of event durations with exponent $\alpha$=1.18 of the best fit power law (same events as in panel A). (**C**) Event sizes scale with event duration to the power $\beta$=2.3. Note that the best fit $\beta$ is not well predicted by crackling noise theory (gold dashed line). (**D**) Each point represents one neural subset (color – power law range) from mouse 1. The neural subsets with strongest correlation to behavior (and largest power-law range) exhibited consistent scaling exponents: $\tau$=0.98 ± 0.11, $\alpha$=1.18 ± 0.19, and fit $\beta$=2.18 ± 0.17. Neural subsets with weaker behavioral correlation (smaller power-law range) had widely varying best fit exponents. Contour lines surround points with power-law range >3.5 decades. (**E**) Summary of all neural subsets with power-law range >3.5 decades from all experiments. Each semi-transparent gray patch corresponds to one contour like the example shown in panel D. The behavioral correlation was chosen to represent the behavior with the greatest correlation for each neural subset.

we analyzed here supports predictions from the theory of criticality. Our next goal is to answer this question.

According to prevailing theories of criticality in neural systems, both the sizes and durations of events should be power-law distributed. Moreover, the theory of crackling noise, which explains a broad class of stick-slip critical phenomena (**Sethna et al., 2001**), predicts that the size power-law exponent $\tau$ and the duration power-law exponent α should be related to how size scales with duration. More specifically, $S(T) \sim T^{\beta}$ where $\beta_{crackling} = (\alpha - 1)/(\tau - 1)$. Many previous observations of scale-free neural activity support this prediction with values of $\beta_{crackling} \approx$ 1.1–1.3 (**Fontenele et al., 2019**; **Fosque et al., 2021**; **Friedman et al., 2012**; **Ma et al., 2019**; **Shew et al., 2015**). To test this theory, we first found the best-fit power-law and its exponent $\tau$ for the size distribution (**Figure 5A**). Then, we fit a power-law to the event duration distributions for the same set of events that were in the

range of power-law scaling for event sizes (*Figure 5B*), thus obtaining α. And finally, we computed a best-fit β by doing a linear fit to log(S) versus log(T) as shown in *Figure 5C*. We found that $\tau$, α, and β varied dramatically across different neural subsets (*Figure 5D*). However, if we examined only the behaviorally-correlated subsets, i.e., those with a large power-law range (*r*>3.5), we found that the power-law exponents were relatively consistent: $\tau$=1.0 ± 0.1, $\alpha$=1.2 ± 0.2, and fit $\beta$=2.2 ± 0.2 (mean ± SD, *Figure 5E*). Thus, we conclude that power-law scaling exists for both sizes and durations of neural events and that size scales with duration according to another power-law. The existence of these scaling laws is consistent with criticality, but we found that our measured values of β were not consistent with crackling noise theory predictions; best fit β was not close to (α - 1)/($\tau$ −1). This was typically because $\tau$ was near one, resulting in small, sometimes negative values of $\tau$ −1 in the denominator of the predicted β formula. Moreover, our observations of β were greater than two, which is not close to previously reported values from experiments – around 1.1–1.3. This suggests that if the scale-free neural activity we observe has its origins in criticality, it may be a type of criticality that is different than previous observations. Indeed, this discrepancy with the existing theory is part of the motivation for the new model we present below. Additional scaling laws from the experiments are compared to the model results below.

Finally, we examined the exponents for distributions of behavioral events. Considering the behaviors with a large power-law range (*r*>3), we found $\tau$=1.2 ± 0.2, $\alpha$=1.4 ± 0.3, and fit $\beta$=1.8 ± 0.4. These exponents were related to the exponents and power-law ranges for neural event distributions, but only if we considered the neural subsets with large power-law range (*r*>3.5 decades) and strong correlation to behavior (*R*>0.6). We found that $\tau$, α, r, and β were significantly correlated for neural and behavioral events (Pearson correlation $R_\tau$ =0.5, p<10$^{-17}$; $R_\alpha$=0.2, p<0.002; $R_r$ = 0.4, p<10$^{-66}$; $R_\beta$=0.3, p<10$^{-4}$). For these comparisons, we paired each neural subset with the type of behavior that was most correlated to that subset.

## Correlations of individual behavioral and neural events

The calculation of correlations between neural subsets and behavior, like those in *Figure 4*, represents the entire recording duration, leaving open several questions. Are the high correlations in *Figure 4* simply due to coarse, on-off dynamics of the neural and behavioral activity, or are there strong correlations at the more detailed level of the faster fluctuations during a single behavioral event? To answer these questions, we computed event-specific correlations – computed between the behavior time series and the neural subset time series during each behavioral event, for each neural subset (*Figure 6A*). For instance, for 1000 neural subsets and 1000 behavioral events (which are typical numbers for this data), we would calculate 1 million correlation values. Each correlation is based on the relatively short time series between the start and stop of the corresponding behavioral event. These event-specific correlations were often quite high (>0.9). However, the chance-level occurrence of such high correlations can also be high for such short time series. We account for such chance-level correlations by repeating the calculations for time-shifted control data (Methods), which defines the event-specific, subset-specific chance-level occurrence rate.

First, we asked how many neural subsets were strongly correlated with each behavioral event and whether this count depended on behavioral event size. We found that many behavioral events had a significant number of such strongly correlated subsets and that the larger behavioral events tended to have a larger number of strongly correlated subsets (*Figure 6B*). Next, we asked how many behavioral events were strongly correlated with each neural subset and whether this count depended on the power-law range of the neural subset. We found that many subsets were strongly correlated with a significant number of behavioral events and that the subsets with greater power-law range tended to be correlated with more behavioral events (*Figure 6C*). The example results in *Figure 6* are shown for all mice and all types of behaviors in *Figure 6—figure supplements 1–2*. Finally, we considered the time during which the mouse was active and asked what fraction of that time had a significant number of strongly correlated neural subsets. This fraction of time was often as high as 0.5 (*Figure 6D*). Thus, we conclude that the detailed, fast fluctuations of behavior are significantly and strongly correlated with scale-free neural activity.

## Anticorrelated neural subsets

How is it possible for the total population to not exhibit scale-free fluctuations, while certain subsets are scale-free? A clue to this mystery is apparent upon close inspection of the results in *Figure 4*.

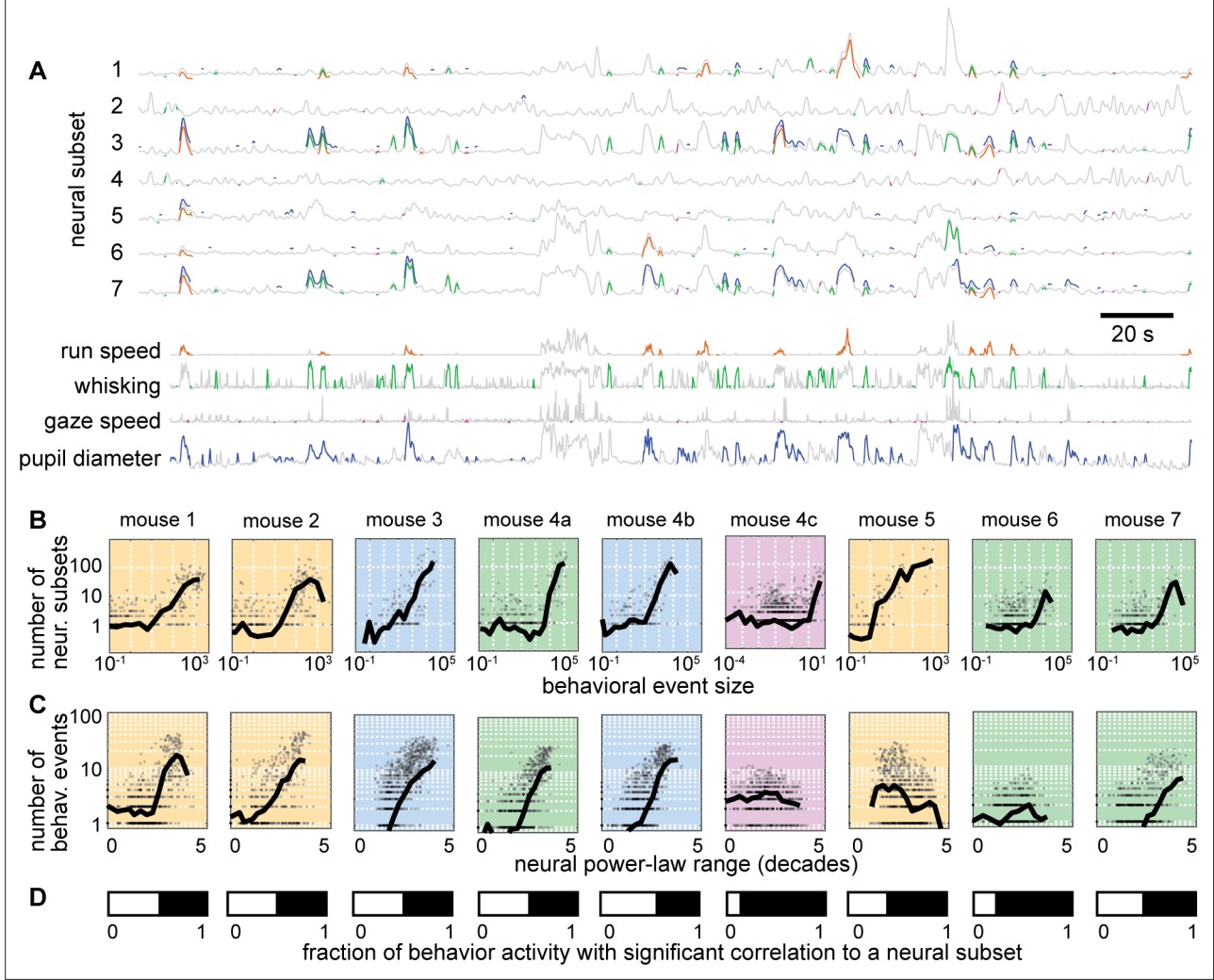

**Figure 6.** Individual neural events correlate with specific behavioral events. (**A**) Time series from 7 example neural subsets and the four behaviors. Each behavioral event with significant correlation to a neural event is indicated with a pair of colored line segments, one on the behavioral time series, one on the neural time series. (**B**) Behavioral events with greater size were strongly correlated with more neural events (i.e. more neural subsets). (**C**) Neural subsets with greater power-law range were significantly correlated with a larger number of behavioral events. For panels B and C, each plot summarizes one recording, with one gray point per behavioral event. Black line is a moving average of the points. Background color indicates type of behavior: orange – running; blue – pupil; green – whisking; purple – gaze. (**D**) The white bar indicates the fraction of behaviorally-active time with strong correlation to a significant number of neural groups. *Figure 6—figure supplements 1–2* show similar results for all recordings and behaviors.

The online version of this article includes the following figure supplement(s) for figure 6:

**Figure supplement 1.** These plots present the same results as *Figure 6B*, but for all mice and all behaviors.

**Figure supplement 2.** These plots present the same results as *Figure 6C*, but for all mice and all behaviors.

Some neural subsets are strongly correlated with behavior, while other subsets are strongly anticorrelated, similar to previous reports from the motor cortex (*Zagha et al., 2015*) and prefrontal cortex (*Garcia-Junco-Clemente et al., 2017*). Since the scale-free subsets are often strongly correlated with behavior, it stands to reason that some neural subsets are strongly anti-correlated with the scale-free subsets. We show in *Figure 7* that this fact explains why the total population does not exhibit scale-free dynamics. When two anticorrelated subsets are averaged together, they cancel out, resulting in relatively small fluctuations at the level of the total population (*Figure 7A*). The broad range of correlations and anticorrelations in each recording are more apparent using 'correlation spectra' (*Figure 7B*). For each previously defined seed neuron, we ranked all the other neurons according to their correlation with the seed neuron. These correlation spectra often reveal large numbers of strongly anticorrelated neurons. To directly show how cancelation of anticorrelated subsets abolishes scale-free

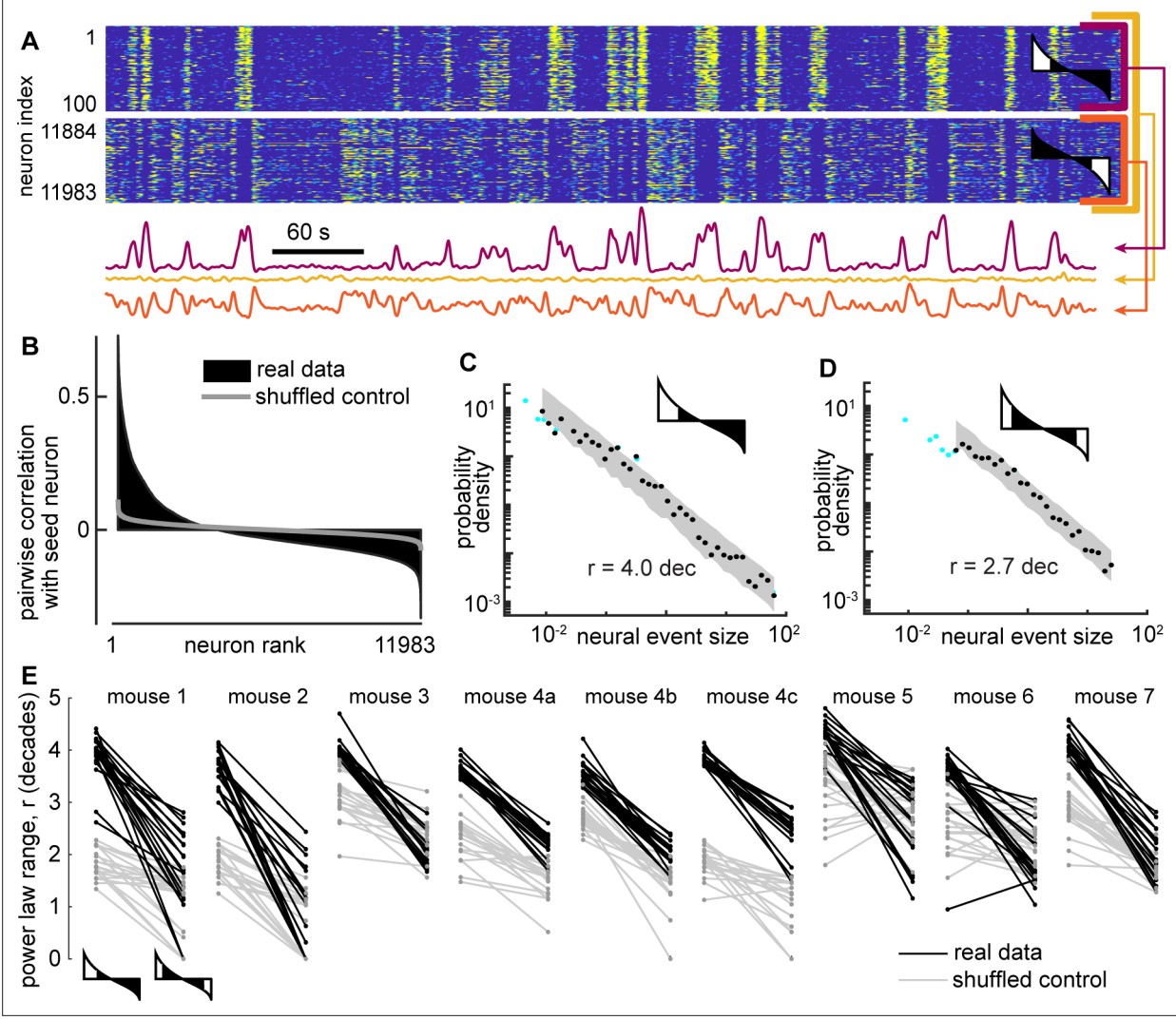

**Figure 7.** Cancelation of anticorrelated neural subsets hide scale-free neural activity. (**A**) Activity of a subset of 100 neurons (top, purple) with large power-law range, another subset of 100 neurons (bottom, orange) that are anticorrelated with first subset, and the total population (yellow). (**B**) Example correlation spectrum, showing the pairwise correlation coefficients between one seed neuron and all other neurons, ranked in descending order. Gray - control spectrum based on randomly time-shifted neurons. (**C**) Example neural event size distribution for a subset based on the top 100 most correlated neurons. (**D**) Example event size distribution for a subset based on the top 50 most correlated neurons and 50 most anticorrelated neurons. Note that the power-law range is greatly reduced compared to panel C. (**E**) The power-law range for the 50 most correlated (left) is much greater than that of the 50 most extreme (right) neurons. Shown are results for the 20 subsets with the greatest power law range for each mouse. Gray lines represent time-shifted controls, which exhibit smaller power-law range and smaller drop in range due to anticorrelated neurons. In panels A, C, D, and E, the cartoon correlation spectra indicate which neurons were included (white) in subsets and which were not (black).

dynamics, we re-computed the power-law range, but this time defined each subset to include the 25 most correlated and the 25 most anti-correlated neurons, relative to each seed neuron. The power-law range for these canceling subsets was greatly reduced compared to the original correlated subsets for all recordings (*Figure 7C–E*).

## Stochastic winner-take-all competition and criticality

What mechanisms might be responsible for scale-free dynamics among a certain subset of neurons that are obscured due to cancelation with anticorrelated subsets of neurons? Theoretical models for studying scale-free dynamics in neural networks have often employed randomly connected (Erdos-Renyi), networks of excitatory and inhibitory neurons, tuned to operate at criticality (*Gautam et al., 2015*; *Li and Shew, 2020*; *Yang et al., 2012*). However, such models do not manifest qualitatively

different slow fluctuations in different subsets of neurons. More specifically, large groups of anticorrelated neurons do not occur in previous models. How can traditional models be changed to account for our observations? Here, we propose that a non-random network structure is needed. Indeed, experiments have shown that connectivity among cortical neurons is far from random (*Cossell et al., 2015*; *Jiang et al., 2015*; *Song et al., 2005*); this non-random structure is particularly pronounced among different types of inhibitory neurons (*Jiang et al., 2015*; *Pfeffer et al., 2013*; *Pi et al., 2013*; *Wall et al., 2016*). To our knowledge, this cell-type-specific connectivity has been ignored in previous models of scale-free neural activity.

Which aspects of this network structure might explain our findings? We initially sought clues from well-known circuit motifs that generate the anticorrelated activity underlying many types of animal locomotion (*Kiehn, 2016*). A common motif in these circuits, sometimes termed 'crossing inhibition' or 'winner-take-all,' entails two excitatory circuit nodes, say e+ and e−, whose activity is anticorrelated because they interact via at least one inhibitory node; e+ excites the inhibitory node, which suppresses e-, and vice versa, e- excites the inhibitory node which suppresses e+ (*Figure 8A–D*). Such circuit motifs certainly are common in the cortex, but they are mixed and interconnected with numerous other motifs. Even Erdos-Renyi networks contain many motifs of this type. Is it plausible that 'winner-take-all' motifs are responsible for our findings?

As shown in *Figure 8A*, we tested this possibility using a computational model of 1000 binary neurons divided into two excitatory groups (e+ and e− with 400 neurons each) and two groups of inhibitory neurons (i+ and i− with 100 neurons each). We considered two inhibitory groups, instead of just one, to account for previous reports of anticorrelations between VIP and SOM inhibitory neurons in addition to anticorrelations between groups of excitatory neurons (*Garcia-Junco-Clemente et al., 2017*). The excitatory groups were densely connected within each group (50% connectivity) and sparsely connected across groups (5%), consistent with experimental observations (*Cossell et al., 2015*; *Song et al., 2005*). We also included dense crossing inhibition according to known connectivity among some inhibitory cell types (*Jiang et al., 2015*; *Pfeffer et al., 2013*; *Pi et al., 2013*; *Wall et al., 2016*). Excitatory and inhibitory synaptic weights were of the same absolute magnitude, but opposite in sign, and the connectivity matrix was normalized such that its largest eigenvalue was one (*Figure 8B*), following previous studies of scale-free dynamics and criticality (*Gautam et al., 2015*; *Larremore et al., 2011*; *Li and Shew, 2020*). Each neuron fired probabilistically, in proportion to the sum of its inputs (Methods).

We found that the e+ and e− neural subsets in this model produced large, slow, scale-free fluctuations (*Figure 8C*) like those observed in the scale-free subsets of neurons from our experimental observations (*Figure 7A*). The dynamics of e+ were strongly anticorrelated with those of e−, and i+ was anti-correlated with i− (*Figure 8C*), resulting in cancelation and a lack of scale-free dynamics when the entire population was considered together (*Figure 8D*). All neurons in the model were driven by unstructured random input; there was no statistical difference between the input to different subsets. Thus, the anticorrelated switching behavior in our model emerges stochastically. These results were robust to a wide range of different input rates (*Figure 8E*), provided that the largest eigenvalue of the connectivity matrix was near 1. If the largest eigenvalue was increased beyond one, the dynamics stopped switching between e+ and e−, instead getting locked into either e+ active and e− inactive or vice versa (this occurred to the right of the red dashed lines in *Figure 8E and F*). Thus, the stochastic switching between e+ and e− activity is a type of critical dynamics occurring at a boundary (red dashed line) in the model parameter space. Such critical winner-take-all dynamics never actually 'choose a winner,' instead switching randomly between the two 'equally deserving winners.' Scale-free anticorrelated fluctuations were also abolished when the density of connections between e+ and e− was increased beyond about 10% (*Figure 8F*). Above this density of connectivity between e+ and e−, the network effectively behaves like a single network, like previous models of criticality and scale-free dynamics.

In addition to testing dependence on the largest eigenvalue, the input rate, and the density of e+-to-e− connectivity, we did a systematic exploration of different network structures. We sought to understand if the network structure considered in *Figure 8A–F* is the only structure that reproduces our experimental observations, or if some other network structures could also work. *Figure 8G* summarizes the results of testing several other network structures. We found that crossing inhibition mediated by a single inhibitory population could suffice. The disinhibitory motif highlighted by other

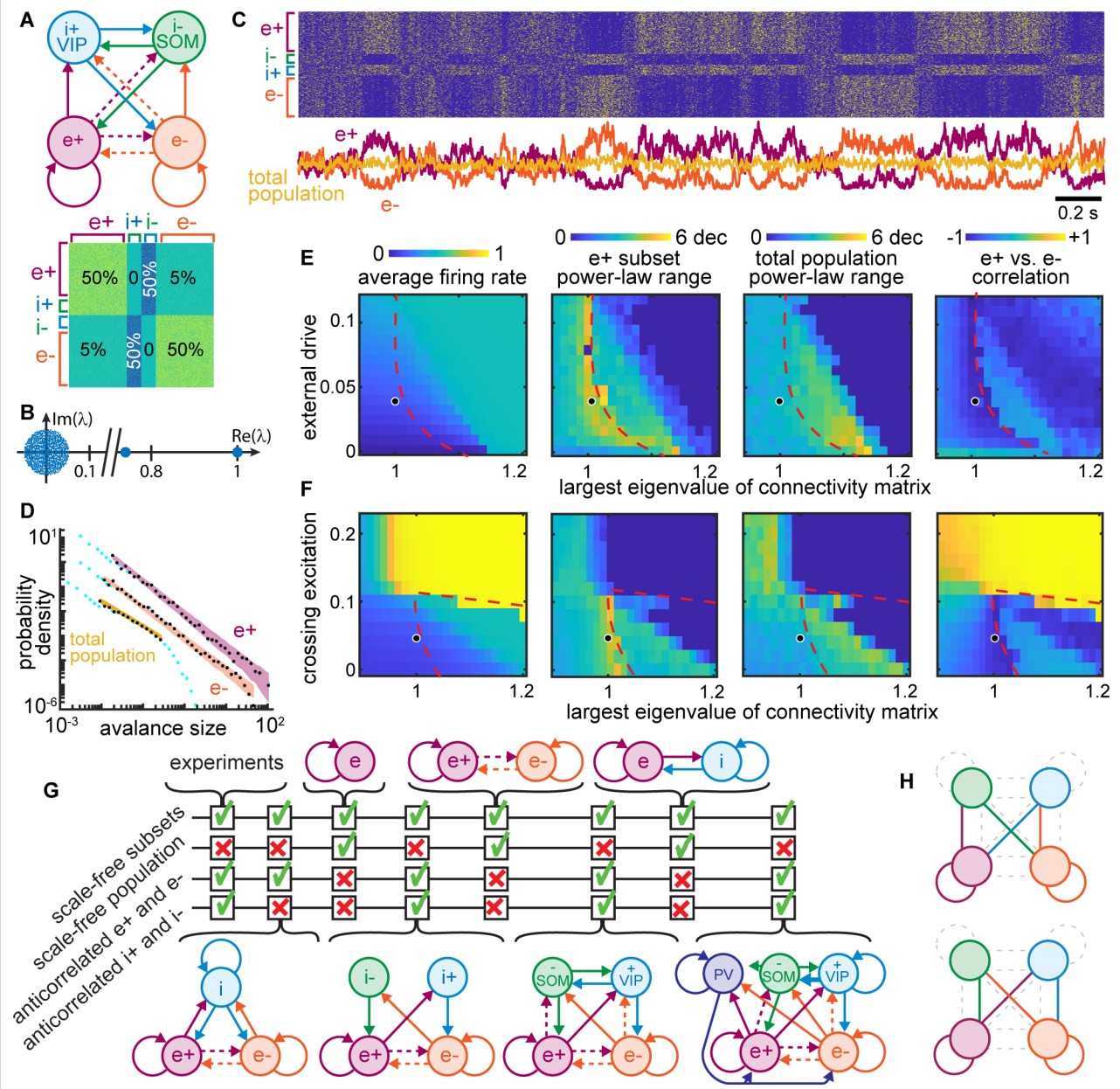

**Figure 8.** Winner-take-all network structure explains scale-free anticorrelated subsets. (**A**) Connectivity diagram (top) and matrix (bottom) illustrate interactions among four populations. Solid, dashed lines indicate dense (50%), sparse (5%) connectivity, respectively. Two excitatory groups (e+ and e−) inhibit each other via dense crossing inhibition. (**B**) Eigenvalue spectrum of connectivity matrix has two outlying real eigenvalues, the largest equal to 1. (**C**) Timeseries and raster reveal strong anti-correlations between e+ and e− populations which cancel out; total population has small fluctuations. (**D**) Neural event distributions for e+ and e− exhibit large power-law range, while the total population does not. (**E**) Parameter space for model. Model agrees with experiments near dashed line for a wide range of input rates, most with largest eigenvalue is near 1. (**F**) Dense crossing excitation from e+ to e− abolishes winner-take-all dynamics. (**G**) Crossing inhibition motif is important for matching our experimental observations. Realistic topologies, including known connectivity among PV+, VIP, and SOM inhibitory neurons (right), contain this crossing inhibition motif and match our observations, but simpler networks can match as well. (**H**) Considering 8,73,000 different network configurations, only 31 match our experimental observations. All matching configurations had one of these two motifs. Dashed lines indicate unnecessary connections. *Figure 8—figure supplement 1* shows all matching configurations.

The online version of this article includes the following figure supplement(s) for figure 8:

**Figure supplement 1.** We tried 8,73,000 different network configurations.

**Figure supplement 2.** Model with asynchronous population.

**Figure supplement 3.** Single-neuron variance and pairwise covariance are larger for neural subsets with large power-law range.

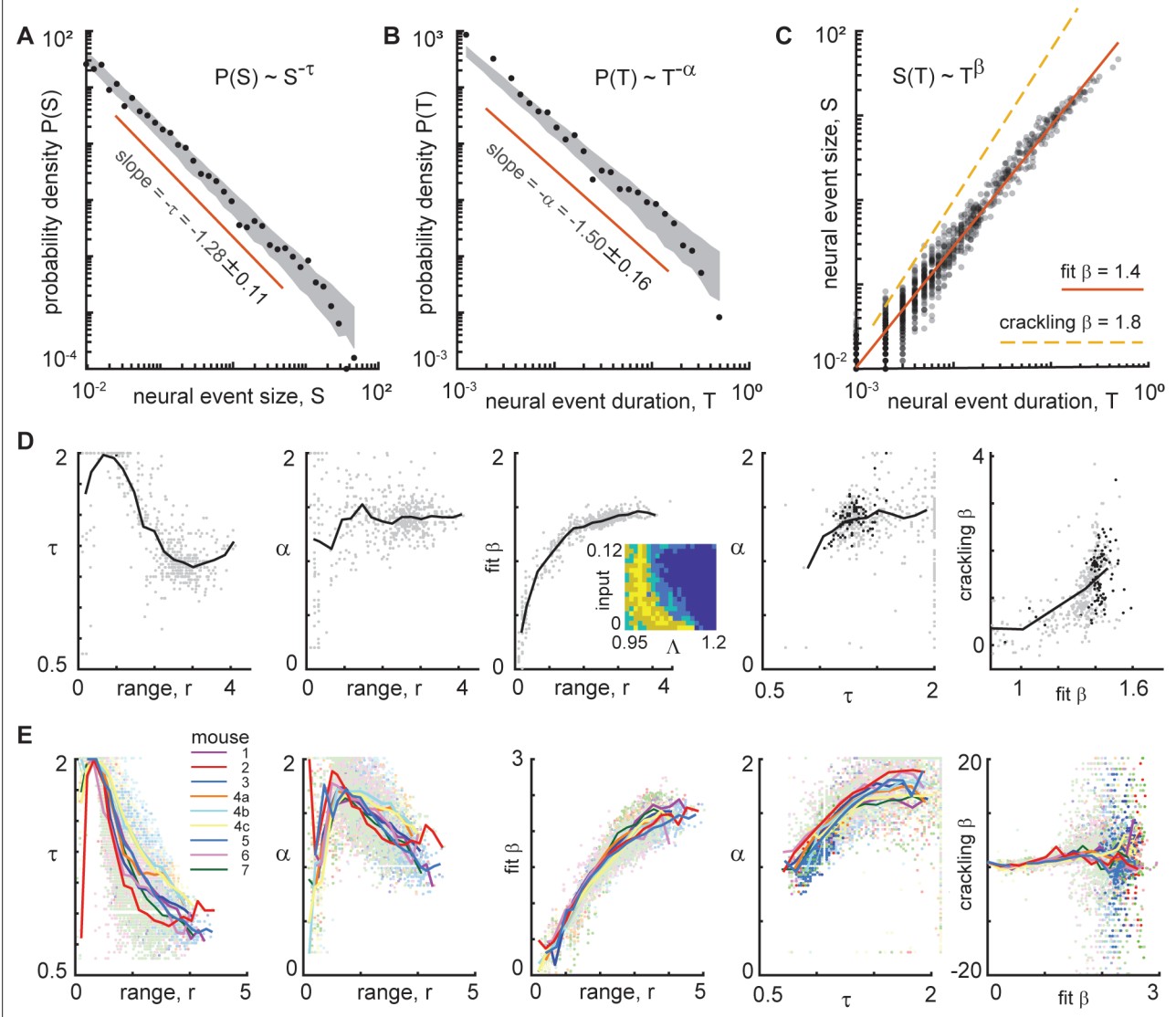

**Figure 9.** Consistent scaling laws for large power-law range in model and experiment. (**A**) Model example neural event size distribution with exponent $\tau$=1.28 of the best fit power law. (**B**) Distribution of event durations from model with exponent $\alpha$=1.5. (**C**) Event sizes scale with event duration to the power $\beta$=1.4. Note that the best-fit $\beta$ is not well predicted by crackling noise theory (gold dashed line). (**D**) For the model, best fit scaling exponents $\tau$, $\alpha$, and $\beta$ are consistent for large power-law range, but not for low power law range. Each gray point represents one run of the model. Different points come from different parameters (input and $\Lambda$) with non-zero power-law range. Inset in middle shows how many points came from each parameter combination (yellow = 4, dark blue = 0). Black points in right panels represent model parameters with power-law range above three decades. (**E**) Same as panel D, but experimental data are shown. Each point represents one neural subset. Color represents mouse. Experiment and model follow similar trends, but experiments tended to have smaller $\tau$ and larger $\beta$ than the model, for large r. Crackling noise predictions of $\beta$ are poor for both experiment and model. Dark colored points in right panels represent neural subsets with power-law range above 3.5 decades.

recent studies (*Garcia-Junco-Clemente et al., 2017*) is, thus, not necessary to reproduce our observations. Importantly, we showed that our observations can also occur in a more realistic network including a third population of inhibitory neurons with connectivity like that of PV neurons, according to known connectivity (*Jiang et al., 2015*; *Pfeffer et al., 2013*). We employed a brute force search considering 873,000 different combinations of dense, sparse, or zero connectivity for the 16 connections among the four network nodes in *Figure 8A* (more detail in *Figure 8—figure supplement 1*). We found only 31 configurations that matched the four criteria in *Figure 8G*. All matching configurations included one of two dense crossing inhibition motifs as well as segregated excitatory groups e+ and e− (*Figure 8H* and *Figure 8—figure supplement 1*). In fact, these two motifs were sufficient for predicting which configurations match our experimental results with 80% accuracy (Methods).

Finally, we considered the exponents and scaling relationships for the event size and duration distributions from the (example shown in *Figure 9A, B and C*). We considered the parts of model parameter space with non-zero power-law range for e+ (*Figure 9D* inset). We compared to experimental results from all neural subsets and all recordings (*Figure 9E*). For smaller power-law ranges, $\tau$ and α were more scattered and inconsistent. For the model, we found $\tau$=1.3 ± 0.12, $\alpha$=1.4 ± 0.15, and $\beta$=1.4 ± 0.04, for power-law range above three decades. For both the experiment and the model, the best-fit β was typically not well predicted by crackling noise theory; a confirmation of crackling noise theory would have the points falling on the unity line in the right panels of *Figure 9D and E*.

While our model offers a simple explanation of anticorrelated scale-free dynamics, its simplicity comes with limitations. Perhaps the most obvious limitation of our model is that it does not include neurons with weak correlations to both e+ and e− (those neurons in the middle of the correlation spectrum shown in *Figure 7B*). In *Figure 8—figure supplement 2*, we show that our model can be modified in a simple way to include asynchronous neurons. Another limitation is that we assumed that all non-zero synaptic connections were equal in weight. We loosen this assumption allowing for variable weights in *Figure 8—figure supplement 2*, without changing the basic features of anticorrelated scale-free fluctuations. Future work might improve our model further by accounting for neurons with intermediate correlations, i.e., less sharp boundaries of the e+ and e− groups.

## Discussion

We have shown that ongoing, untrained locomotion, whisking, and pupil diameter changes in mice often exhibit scale-free dynamics. This scale-free behavior is directly related to concurrent scale-free cortical neural activity among certain subsets of neurons, with a strong one-to-one correspondence between many behavioral and neural events. Moreover, the subsets of neurons with the greatest range of scale-free-ness (i.e. the greatest power-law range) were the most correlated with behavior.

Why might a large power-law range among neurons be important for behavior? The behavioral variables we analyzed are likely to involve both local interactions among neighboring neurons and long-range coordination across many brain regions. For example, volitional movements like running and whisking presumably require coordinated interactions within and among motor, sensory, and other brain regions associated with decisions to move. One way to manifest such multiscale coordination from local to brain-wide circuits is to have neural events with sizes that occur with diverse scales, i.e., neural events with large power-law range.

Previous studies suggest additional functional implications of this close relationship between scale-free behavior and brain activity. First, scale-free neural activity has been associated with multiple functional benefits for sensory information processing and memory (*Clawson et al., 2017*; *Gautam et al., 2015*; *Shew and Plenz, 2013*; *Shriki and Yellin, 2016*; *Suárez et al., 2021*), while scale-free behavior has been associated with benefits for foraging, search, and decision making (*Abe, 2020*; *Barabási, 2005*; *Garg and Kello, 2021*; *Sims et al., 2008*; *Sorribes et al., 2011*; *Viswanathan et al., 1999*; *Wosniack et al., 2017*). Our results suggest that these two lists of functional benefits, previously considered separately, may in fact be a single unified list that occurs together. This long list of benefits may explain why several types of neural plasticity have been shown to maintain scale-free neural activity (*Hoseini et al., 2017*; *Ma et al., 2019*; *Shew et al., 2015*).

What mechanisms are responsible for scale-free neural activity? The prevailing view is the criticality hypothesis (see for example *Beggs and Plenz, 2003*; *Muñoz, 2018*; *Shew and Plenz, 2013*; *Wilting and Priesemann, 2019*). In this view, scale-free dynamics occur because the system operates near criticality, i.e., near the tipping point between two distinct dynamical regimes - one synchronous, the other asynchronous. However, previously studied models of criticality fail to explain several crucial components of our observations. First, we observed that scale-free activity is limited to subsets of neurons, which were strongly anticorrelated with other subsets of neurons. This apparent stochastic winner-take-all competition between subsets results in the cancelation of fluctuations at the level of the total population, thus obscuring the scale-free dynamics of the subsets. Previous models also do not agree with the power-law scaling relationships we found – especially our measurements of β in the relation S(T)~T$^{\beta}$. Our results suggest that a new type of criticality may be required to explain these observations. One possibility suggested by our model is that the scale-free dynamics we observe occur at the boundary between winner-less switching and single-winner locked-in dynamics (the red

dashed line in *Figure 8E and F*). Additional theoretical efforts are necessary to more fully explore how the traditional criticality hypothesis relates to the competitive criticality suggested by our model.

At first glance, our results appear to not only contradict existing theory of criticality in neural systems, but also prior experimental observations based on different measurement methods. Most prominently, many previous studies have observed scale-free neural activity based on fast electrophysiological measurements using the total measured population, without accounting for anticorrelated subsets of neurons. Can fast ephys dynamics be reconciled with the slow calcium imaging signals we analyzed here? A detailed reconciliation of these different timescales of measurement will require more extensive investigation, but we provide a partial answer here (*Figure 3—figure supplements 1–2*). We analyzed an electrophysiological recording of spike times from 294 single units, also reported in the *Stringer et al., 2019* paper. We found that when the activity is analyzed at a fast time scale (5 ms resolution), scale-free activity is clear using the full population and anticorrelations were very weak. The exponent of the power law was near –2, consistent with some previous reports based on time-resolved spike recordings in awake animals (*Fontenele et al., 2019*; *Ma et al., 2019*). In contrast, when the activity was analyzed at a slow time scale (330 ms resolution) like the calcium imaging data, strong anticorrelations emerged and scale-free dynamics were much more apparent (i.e. power law range was greater) when analyzed in subsets of neurons. This result (*Figure 3—figure supplement 2*) suggests that our findings are not inconsistent with prior studies of scale-free activity based on time-resolved electrophysiological data and calls for a more thorough study of how correlation structure and scale-free dynamics differ and relate across spatiotemporal scales. At the coarse time resolution of the calcium imaging data, we note that the neural subsets with large power-law range tended to be composed of neurons with high activity variance and high pairwise covariance (*Figure 8—figure supplement 3*). We note that a similar point about the emergence of anticorrelations at slow timescales was made in the original *Stringer et al., 2019* paper.

What causes the scale-free fluctuations in behavior we observed from experiments? From the point of view of our model and the criticality hypothesis more generally, scale-free-ness originates in the dynamics of neural circuits. Indeed, the model generates scale-free fluctuations without any interactions with an 'outside world.' These internally generated scale-free neural dynamics then, might manifest as scale-free behavior. However, recalling that the experimental measurements reported here were from primary visual cortex, one might ask whether the scale-free fluctuations we observed in V1 are driven by scale-free fluctuations in visual input. In our view, this is unlikely given several facts. First, the gaze motion often had the least convincing scale-free-ness in our results (i.e. smallest power-law range, *Figure 2C*) and gaze motion would be the most obvious reason for changes in visual input (there were no dynamic visual stimuli on a viewed screen for these recordings). Second, the Stringer et al., paper (2019), from which our data originated, showed that spatially distributed electrophysiological recordings from multiple brain regions (not just sensory) were correlated to behavior in a similar way. This suggests that our results here may be a brain-wide phenomenon, not just a V1 phenomenon.

Setting aside speculations, our results firmly establish that complex fluctuations and events that make up scale-free neural activity are not 'background noise' nor 'internal' cognitive processes as is often supposed. Our findings call for a revision of these traditional views; scale-free neural activity directly causes (or perhaps is caused by) scale-free behavior and body motion.

## Materials and methods

**Key resources table**

| Reagent type (species) or resource | Designation | Source or reference | Identifiers | Additional information |
| --- | --- | --- | --- | --- |
| Software,algorithm | Matlab | Mathworks | R2018a | |

### Animals and data acquisition

The data analyzed here were first published in *Stringer et al., 2019* and are publicly available at doi:10.25378/janelia.6163622.v4.

All animal protocols and data acquisition methods are described in the original publication. All experimental procedures were conducted according to the UK Animals Scientific Procedures Act (1986). Experiments were performed at University College London under personal and project licenses released by the Home Office following appropriate ethics review. In brief, multiplane calcium imaging

during periods with no visual stimulation was performed in seven adult mice (P35 to P125) bred to express GCaMP6s in excitatory neurons. Our labeling of the different recordings corresponds to the original authors' labeling system of the data as follows:

mouse 1 - spont_M150824_MP019_2016-04-05,
mouse 2 - spont_M160825_MP027_2016-12-12,
mouse 3 - spont_M160907_MP028_2016-09-26,
mouse 4 a - spont_M161025_MP030_2016-11-20,
mouse 4b - spont_M161025_MP030_2017-06-16,
mouse 4 c - spont_M161025_MP030_2017-06-23,
mouse 5 - spont_M170714_MP032_2017-08-04,
mouse 6 - spont_M170717_MP033_2017-08-18,
mouse 7 - spont_M170717_MP034_2017-08-25.

## Neural data pre-processing

Beginning from the deconvolved fluorescence traces (variable called Fsp in the original data set), first, the data were z-scored. That is, for each neuron, we subtracted its time-averaged activity and divided by its standard deviation. Second, we applied a low-pass filter (cutoff frequency 0.2 Hz, 2nd order butterworth filter, using Matlab filtfilt function).

## Definition of neural events

From previous studies of scale-free neural activity, there are three established strategies for defining neural events. The original approach was to first bin time (with time bin durations usually defined by the average inter-spike-interval, or average interval between LFP peaks) and define an event or avalanche as a sequence of time bins with at least one spike or LFP peak, preceded and terminated by an empty time bin (*Beggs and Plenz, 2003*). This approach has been very useful when the data come from a relatively small number of neurons or electrodes but runs into difficulty with very large numbers of single neurons. The simple reason for this difficulty is that for very large numbers of neurons, there is simply no empty time bins (down to the time resolution of measurements), resulting in a single event that never ends. A second approach that better accounts for the possibility of more than one event occurring at the same time came from analyzing fMRI brain activity (*Tagliazucchi et al., 2012*). This approach assumes that an event must evolve in a spatially contiguous way as it spreads through the brain. This approach, while good for spatially coarse measurements like fMRI, does not make sense for a small local circuit with a single neuron resolution like that analyzed here. The third approach is the one we adopted here, described in our Results section, which is the only good option, to our knowledge, for single neuron resolution and very large numbers of neurons.

## Power-law fitting and range

To assess the scale-freeness of behavioral events and neural population events, we developed an algorithm for finding the range of event sizes that are well fit by a power law. The algorithm is based on a maximum likelihood fitting procedure as established in previous work (e.g. *Clauset et al., 2009*; *Langlois et al., 2014*; *Shew et al., 2015*). We fit the measured event size distribution with a truncated power-law with minimum event size $s_{min}$ and maximum event size $s_{max}$, excluding data that fell outside these bounds during the fit process. Similar to previous studies, our algorithm has two fitting parameters - $s_{min}$ and the power-law exponent. $s_{max}$ was not a fitting parameter; it was chosen to be the largest observed event size (for subsets with power-law range >3, the mean ± SD for $s_{max}$ was 59 ± 26 s for event durations and 102 ± 48 for event sizes). We tested exponents between 0.7 and 2 in steps of 0.02. We tested $s_{min}$ values between the smallest observed event size and the largest observed event size, increasing in 10 logarithmically spaced increments per decade. Thus, the power-law range reported in the manuscript has a resolution of 0.1 decades. Importantly, our fitting algorithm is entirely independent of any choice of binning that might be used to visualize the event size distribution.

For our purposes of measuring power law range, previously reported algorithms required improvements to account for (exclude) confounding outlier event sizes. These 'outliers' were rare events that caused noise in the extremes of the distribution tail or head and, in some cases, caused spuriously large power-law range estimates. Accounting for these outliers provides a more conservative estimate

of power-law range. We defined outliers by first ranking all event sizes from smallest to largest. Next, we computed differences in sizes (in decades) for consecutive sizes in this ranked list. Outliers have a large difference in size compared to the following size (or preceding size). We define outliers as events with a size difference greater than 3% of the total range in decades. We also tried 6% as an outlier threshold and found that our results were not very sensitive to this choice (**Figure 2C**).

The fitting algorithm executed the following steps. First, outliers were excluded. Second, events with size less than $s_{min}$ were excluded. Third, the maximum likelihood power-law exponent was calculated. Fourth, we assessed the goodness-of-fit. We repeated these four steps for all the possible $s_{min}$ values, in the end, identifying the largest range (smallest $s_{min}$) that passed our goodness-of-fit criterion.

The goodness-of-fit criterion we adopted here was also new, to our knowledge, and better suited to our goals of assessing power-law range, compared to previous methods. The steps for quantifying goodness-of-fit were as follows. First, we created a cumulative distribution function (CDF) of the real data. Second, we created 500 surrogate data sets drawn from the best-fit truncated power-law. Third, we created 500 CDFs, one for each of the surrogate data sets. Fourth, we resampled all 501 CDFs with 10 logarithmically spaced points per decade, linearly interpolated. Fifth, we calculated the fraction F of points in the resampled CDF of the real data that fell within the bounds of the 500 resampled surrogate CDFs. F is our goodness-of-fit measure. $F=1$ means that the entire range of the real data falls within the expected variation for a perfect power-law, given the number of samples in the data. We explored various goodness-of-fit criteria between $F=0.75$ and $F=0.9$ for the behavioral data (**Figure 2C**) and the neural data (**Figure 3E**). This is a rather conservative goodness-of-fit test, much more conservative than the more typically used Kolmogorov-Smirnov statistic for example.

The 500 surrogate data sets were also used for the purpose of creating the gray-shaded regions in each plotted distribution of avalanche sizes and durations (**Figure 2C**, **Figure 5A and B**, **Figure 9A and B**). First, each surrogate data set was used to create a probability density function (PDF) with the same bins as the real data. Each of these PDFs was slightly different due to statistical noise (i.e. finite sample size). The gray-shaded region represents the range between which 90% of the surrogated PDFs fell. Thus, the gray region indicates how much variability one should expect for a perfect power law with the best fit exponent and the same range and sample size as the real data. We also use the upper and lower bounds of the gray region to estimate error bars for our exponents (as reported in **Figure 2C**, **Figure 5A and B**, **Figure 9A and B**) We calculate $\tau_{up}$ from the slope of the upper boundary of the gray region and $\tau_{lo}$ from the slope of the lower boundary of the gray region. Then we report $\tau = \tau_{MLE} \pm \tau_{VAR}$, where $\tau_{MLE}$ is the maximum likelihood exponent and $\tau_{VAR} = (\tau_{up} - \tau_{lo})/2$. Considering the event size exponents for all neural subsets with power-law range greater than 3, we found that $\tau_{VAR}=0.07 \pm 0.05$ (mean, SD); corresponding event duration exponents had $\tau_{VAR}=0.11 \pm 0.07$. For all behavioral event size distributions, $\tau_{VAR}=0.19 \pm 0.17$.

In **Figure 2—figure supplement 2**, we report the results of a benchmarking analysis in which we ran our power-law fitting algorithm on synthetic data sets drawn from known power-law distributions. We found that the exponents and power-law ranges found by our fitting algorithm are robust for a reasonable range of outlier exclusion thresholds (including 3% and 6%), different exponents, and sample sizes.

## Statistical significance of correlations

*Figure 6* is based on computing correlation coefficients of short time series of behavior and neural activity. The number of such correlation coefficients computed for each behavioral event was 1000 because there were 1000 neural subsets. In the manuscript, we concluded that a significant number of these correlations were strong. This conclusion was arrived at by comparing to time-shifted control neural data (same as described for the time-shifted controls in the manuscript for *Figure 3*). The time shifts were randomly chosen from the interval 0 s up to the entire duration of the recording. A cyclic permutation was performed such that a time shift of + N samples would move the ith sample to become the $(i+N)^{th}$ and the last N samples become the first N. For each behavioral event, we obtained 1000 time-shifted control correlation coefficients. We then counted how many of the real correlation coefficients were greater than 999 of the time-shifted control correlation coefficients. By chance, we should expect this count to be one. We concluded that there was a significant number of strong correlations if this count was greater than four. A similar reasoning was used to conclude that there were a significant number of strongly correlated behavioral events for each neural subset.

## Computational model

The model consisted of N=1000 binary neurons; the state of the ith neuron at time t is $s_i(t) = 0$ (quiescent) or $s_i(t) = 1$ (firing). The population was divided into four groups called e+, e−, i+, and i−, as diagramed in **Figure 8A**. The e+ and e− groups include 400 excitatory neurons each; i+ and i− include 100 inhibitory neurons each. The dynamics of the neurons are updated synchronously according to

$$s_i(t) = \begin{cases} 1 & \text{with probability } p_i(t) \\ 0 & \text{with probability } 1 - p_i(t) \end{cases}$$

where the activation probability is

$$p_i(t) = f\left(\eta + \sum_{j=1}^{N} C_{ij} s_j(t-1)\right)$$

Here, $\eta$ is a constant representing input from outside the model network and/or a tendency to fire spontaneously without input. We characterized how changes in $\eta$ affect model dynamics in **Figure 8E** (vertical axis, labeled 'external drive'). The connection from neuron j to neuron i is given by $C_{ij}$, one element of the connection matrix depicted in **Figure 8A**. Certain pairs of groups were sparsely connected, while other pairs of groups were densely connected, or not connected at all. 50% of connections (randomly chosen) were zero for two densely connected groups. 95% of connections (randomly chosen) were zero for a sparsely connected pair of groups. All non-zero excitatory connections were set to the same constant c; all non-zero inhibitory connections were set to -c. The value of c was set by first normalizing the entire connection matrix by its largest eigenvalue and then multiplying by another constant $\Lambda$ to obtain the desired largest eigenvalue. We studied how the model dynamics depend on $\Lambda$ in **Figure 8F** and **Figure 8G** (horizontal axis). For the shotgun search of many different circuit configurations, we set $\Lambda = 1$. We also studied how the density of connectivity between e+ and e− impacted model dynamics in **Figure 8F** (vertical axis, called 'crossing excitation').

All model data analysis was done on the population activity and averaged over neurons for each group.

We did a shotgun search for model configurations that result in dynamics consistent with our experimental results. To do this, we first generated a list of all possible configurations of the 16 pairwise group connections. We considered three possible densities for each of these connections: disconnected (0%), sparse (5%), or dense (50%). Thus, the list of all possible connection configurations includes $3^{16}$ (more than 43 million) possibilities. Clearly testing all these possibilities would take a long time, so we pared down the list with the following constraints, which exclude several very unrealistic possibilities:

1. Both e+ and e− groups must have non-zero within-group connectivity.
2. No disconnected components were allowed. Every group had to be reachable by at least one connection.
3. There must be at least one excitatory to inhibitory connection.
4. There must be at least one inhibitory to excitatory connection.

After applying these constraints, the list of possible configurations still included 18,576,000 possible circuits. We tested 8,73,000 of these possibilities, chosen at random. We considered a circuit to be consistent with experimental results if it met the following conditions:

1. Correlation coefficient of e+ and e− must be less than −0.5
2. Correlation coefficient of i+ and i− must be less than −0.5
3. Power law range of e+ or e− must be greater than 3.5 decades
4. Power law range of the total population must be less than 2 decades

We found that only 31 circuit configurations met these conditions. These matching networks are illustrated in **Figure 8—figure supplement 1**. As noted in the main text, all matching configurations featured two structural motifs. First, they included dense self-connections within e+ and within e−, but weak or non-existent connections between e+ and e−. Second, all matching configurations included one of two dense crossing inhibition configurations (**Figure 8H**). Moreover, we found that these two conditions were sufficient to predict which networks would match the experimental results with 80% accuracy if we loosened the criteria for matching experiments slightly. These looser criteria were:

1. Correlation coefficient of e+ and e− must be less than –0.5
2. Correlation coefficient of i+ and i− must be less than –0.2
3. Power law range of e+ or e− must be at least 1.4 times that of the total population

## Additional information

### Funding

| Funder | Grant reference number | Author |
|---|---|---|
| National Institutes of Health | NIHR15NS116742 | Woodrow L Shew |
| National Science Foundation | NSF1912352 | Woodrow L Shew |

The funders had no role in study design, data collection and interpretation, or the decision to submit the work for publication.

### Author contributions

Sabrina A Jones, Conceptualization, Software, Formal analysis, Investigation, Methodology, Writing – review and editing; Jacob H Barfield, Software, Formal analysis, Investigation, Methodology, Writing – review and editing; V Kindler Norman, Software, Formal analysis, Investigation; Woodrow L Shew, Conceptualization, Software, Formal analysis, Supervision, Funding acquisition, Investigation, Methodology, Writing – original draft, Project administration, Writing – review and editing

### Author ORCIDs

Woodrow L Shew ⓘ http://orcid.org/0000-0003-0679-1766

### Ethics

All experimental procedures were conducted according to the UK Animals Scientific Procedures Act (1986). Experiments were performed at University College London under personal and project licenses released by the Home Office following appropriate ethics review.

### Decision letter and Author response

Decision letter https://doi.org/10.7554/eLife.79950.sa1
Author response https://doi.org/10.7554/eLife.79950.sa2

## Additional files

### Supplementary files

• MDAR checklist

### Data availability

The data analyzed here were first published in *Stringer et al., 2019* and are publicly available on figshare at https://doi.org/10.25378/janelia.6163622.v6. Analysis code is publicly available on figshare at https://doi.org/10.1101/2021.05.12.443799.

The following dataset was generated:

| Author(s) | Year | Dataset title | Dataset URL | Database and Identifier |
|---|---|---|---|---|
| Shrew W | 2023 | Code for computational model, power law fitting, power law range [Shew Lab] | https://doi.org/10.6084/m9.figshare.21954389.v1 | figshare, 10.6084/m9.figshare.21954389.v1 |

The following previously published dataset was used:

| Author(s) | Year | Dataset title | Dataset URL | Database and Identifier |
|---|---|---|---|---|
| Stringer C, Pachitariu M, Reddy C, Carandini M, Harris KD | 2018 | Recordings of ten thousand neurons in visual cortex during spontaneous behaviors | https://doi.org/10.25378/janelia.6163622.v6 | Figshare, 10.25378/janelia.6163622.v6 |

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
