## [Editor Report]

This paper is an important study that is of interest to neuroscientists studying the organization of neural activity and behavior. The authors present compelling evidence to link the apparently scale-free distributions of behavioral metrics with scale-free distributions of neural activity. They then explore computationally mechanistic models that could account for these observations. The simulations of mechanistic models are provocative and suggest interesting network-connectivity hypotheses to test in future experiments.

---

## [Decision Letter]

**Decision letter after peer review:**

Thank you for submitting your article "Scale-free behavioral dynamics directly linked with scale-free cortical dynamics" for consideration by *eLife*. Your article has been reviewed by 3 peer reviewers, and the evaluation has been overseen by a Reviewing Editor and Ronald Calabrese as the Senior Editor. The following individual involved in review of your submission has agreed to reveal their identity: Keith B Hengen (Reviewer #1).

The reviewers have discussed their reviews with one another, and although there were some concerns, there was general enthusiasm for the approach and for the ideas presented in the work. Accordingly, the Reviewing Editor has drafted this to help you prepare a revised submission.

Essential revisions:

1. The reviewers would like to see a detailed and thoughtful reflection on the role that 3 Hz Ca imaging might play in the conclusions that the authors derive. While the dataset in question offers many neurons, this approach is, from other perspectives, impoverished – calcium intrinsically misses spikes, a 3 Hz sampling rate is two orders of magnitude slower than an action potential, and the recordings are relatively short for amassing substantial observations of low probability (large) avalanches. The potential concern is that some of this disconnect may reflect optophysiological constraints. One argument against this is that a truly scale free system should be observable at any temporal or spatial scale and still give rise to the same sets of power laws. This quickly falls apart when applied to biological systems which are neither infinite in time nor space. As a result, the severe mismatch between the spatial resolution (single cell) and the temporal resolution (3 Hz) of the dataset, combined with filtering intrinsic to calcium imaging, raises the possibility that the conclusions are influenced by the methods.

2. Another reservation expressed by the referees has to do with the generality of the conclusions drawn from the mechanistic model. One of the connectivity motifs identified appears to be i+ to e- and i- to e+, where potentially i+/i- are SOM and VIP (or really any specific inhibitory type) cells. The specific connections to subsets of excitatory cells appear to be important (based on the solid lines in Figure 8). This seems surprising: is there any experimental support for excitatory cells to preferentially receive inhibition from either SOM or VIP, but not both? More broadly, there was concern that the neat diagrams drawn here are misleading. The sample raster, showing what appears to be the full simulation, certainly captures the correlated/anti-correlated pattern of the 100 cells most correlated with a seed cell and 100 cells most anti-correlated with it, but it does not contain the 11,000 cells in between with zero to moderate levels of correlation. We probably expect that the full covariance matrix has similar structure from any seed (see Meshulam et al. 2019, PRL, for an analysis of scaling of coarse-grained activity covariance), and this suggests multiple cross-over inhibition constraints, which seem like they could be hard to satisfy. The motifs identified in Figure 8 likely exist, but I am left with many questions of what we learned about connectivity rules that would account for the full distribution of correlations. Would starting with an Erdos-Renyi network with slight over-representation of these motifs be sufficient? How important is the homogeneous connection weights from each pool assumption – would allowing connection weights with some dispersion change the results?

3. Putting 2) another way, it's unclear why the averaging is required in the first place. This operation projects the entire population down in an incredibly lossy way and removes much of the complexity of the population activity. Second, the authors state that it is highly curious that subsets of the population exhibit power laws while the entire population does not. While the discussion and hypothesizing about different e-i interactions is interesting, it is possible that there's a discussion to be had on a much more basic level of whether there are topology independent explanations, such as basic distributions of correlations between neurons that can explain the subnetwork averaging. Specifically, if the correlation to any given neuron falls off, e.g., with an exponential falloff (i.e., a Gaussian Process type covariance between neurons), it seems that similar effects should hold. This type of effect can be easily tested by generating null distributions existing code bases. This is an important point since local (broadly defined) correlations of neurons implying the observed subnetwork behavior means that many mechanisms that have local correlations but don't cluster in any meaningful way could also be responsible for the local averaging effect.

4. In general, the discussion of "two networks" seems like it relies on the correlation plot of Figure~7B. The decay away from the peak correlation is sharp, but there does not seem to be significant clustering in the anti-correlation population, instead a very slow decay away from zero. The authors do not show evidence of clustering in the neurons, nor any biophysical reason why e and i neurons are present in the imaging data. The alternative explanation (as mentioned in (b)) is that the there is a more continuous set of correlations among the neurons with the same result. In fact one of the reviewers tested this themself using code to generate some data with the desired statistics, and the distribution of events seems to also describe this same observation. Obviously, the full test would need to use the same event identification code, and so it is quite important that the authors consider the much more generic explanation for the sub-network averaging effect. We recommend assessing the possibility that broader explanations, e.g., in the form of the distributions of correlations accounts for the observed phenomenon. Even with 10K neurons, there are many other forces at play influencing the observed network and while it is nice that e-i networks are one explanation, much less constraining explanations that are still biophysically feasible should be discussed and compared against. I have provided one possible approach (see PDF for code and example figure: https://submit.elifesciences.org/*eLife*_files/2022/05/19/00107349/00/107349_0_attach_6_474_convrt.pdf).

5. Another important aspect here is how single neurons behave. It was not clear if single neurons were stated to exhibit a power law. If they do, then that would help in that there are different limiting behaviors to the averaging that pass through the observed stated numbers. If not, then there is an additional oddity that one must average neurons at all to obtain a power law. We recommend the authors show the full curve so that the readers get a more detailed sense of how averaging effects the power-law interpretation of the data.

6. There is something that seems off about the range of \β values inferred with the ranges of \tau and $\α$. With \tau in [0.9,1.1], then the denominator 1-\tau is in [-0.1, 0.1], which the authors state means that \β (found to be in [2,2.4]) is not near \β_{crackling} = (\α-1)/(1-\tau). It seems as this is the opposite, as the possible values of the \β_{crackling} is huge due to the denominator, and so \β is in the range of possible \β_{crackling} almost vacuously. Was this statement just poorly worded?

7. It is not clear if there is more to what the authors are trying to say with the specifics of the scale free fits for behavior. Apparently, these results are used to motivate the neural studies, but aside from that, the details of those ranges don't seem to come up again. Given that the primary connection between neuronal and behavioral activity seems to be Figure 4. The distribution of points in these plots seem to be very lopsided, in that some plots have large ranges of few-to-no data points. It would be very helpful to get a sense of the distribution of points that are a bit hard to see given the overlapping points and super-imposed lines. We recommend that the authors add distribution information to the plots in Figure 4B to give a sense of how points are spread through the [correlation with behavior]-by-[power law range] space. Potential plots might be a co-located histogram, or perhaps an uncertainty estimate as a function of correlation based on the number of points and variance. This would help show significance of the curves in a way that accounts for the uneven spread of datapoints.

8. Neural activity correlated with some behavior variables can sometimes be the most active subset of neurons. This could potentially skew the maximum sizes of events and give behaviorally correlated subsets an unfair advantage in terms of the scale-free range. In a similar vain to 8), what are the typical dynamic ranges for subsets correlated and uncorrelated with behavior? We recommend showing a number of these to see if those dynamic ranges are impacting the possible ranges in the [correlation with behavior]-by-[power law range] plots. Perhaps something like curve in each plot showing the minimum maximum value of the power law range per correlation range. In general, the reviewers struggled with the interpretation of Figure 4b in the sense that there seems to be such variability between mice. How much do the authors feel that this is a difference in neural populations imaged, vs changes in imaging conditions (illumination, window clarity, optical alignment) or differences in mouse activity levels?

*Reviewer #1 (Recommendations for the authors):*

This paper feels highly polished and thorough in its presentation. I truly enjoyed reading it and believe it will be of value to the community.

A nuanced question: mouse #5 and mouse #6 consistently seems to break the rules implied by the rest of the dataset. The behavioral descriptors look normal in figure 2, but neural structure is notably different from the rest of the group in 3E, 4B, 6C, 7E, and S4 (that one is behavior). Is there anything meaningfully different about these recordings (n cells, mean event rate, anatomical location etc)?

*Reviewer #2 (Recommendations for the authors):*

Comments, questions, and technical issues for the authors.

1. The values of tau (size exponent) from neural avalanches are surprisingly low. For instance, in the Ma et al. (Hengen) 2019 Neuron paper, tau ranged from 1.5 to 1.9, sometimes more than 2, but never less than 1. I am wondering how the imaging preprocessing affects the estimate of tau. For instance, if you started with simulated spiking activity that has avalanches with size pdf that go as a power law with exponent tau, and then use a forward model to generate "calcium imaging," and then apply deconvolution, z-scoring, and low-pass filtering, and then measure the avalanches again: what is the new tau?

2. This may be an ignorant question (apologies). The power law range is quantified in decibels (dB) throughout the paper; do you actually mean decades?

3. Related to the set-up of the model, was there a reason that there are no adaptation mechanisms in this network model, as there often are in mechanistic models for avalanche criticality (including past work by the authors of this paper, e.g. Shew's 2015 Nature Physics paper)? Also, there appears to have been an error with the reference manager, as this reference shows up twice in the reference list.

4. It would be helpful the authors could elaborate on predictions that their results make for future studies. Maybe this is rather technical, but can you tell us when you expect to find a power-law distribution as a function of how much of the population is sampled and for how long? What if you were analyzing Neuropixels data, where you lose the extensive spatial sampling (and the restriction to pyramidal cells only) but you gain 3 orders of magnitude in temporal resolution?

5. On page 8, you ask "are all behavioral events equally correlated to their concurrent neural events, or are certain neural events from certain subsets of neurons more strongly related to behavioral events?" I don't understand the question. What does it mean for all behavioral events to be equally correlated to concurrent neural events? Aren't "concurrent neural events" in specific subsets of neurons?

*Reviewer #3 (Recommendations for the authors):*

1. Limits of calcium imaging: My recommendation is to assess mathematically the potential impact of missing data on the range and power-law slope estimates, which are the primary values used throughout the paper.

2. Correlations and power-laws in subsets.

2a-c. My recommendation is to assess the possibility that broader explanations, e.g., in the form of the distributions of correlations accounts for the observed phenomenon. Even with 10K neurons, there are many other forces at play influencing the observed network and while it is nice that e-i networks are one explanation, much less constraining explanations that are still biophysically feasible should be discussed and compared against. I have provided one possible approach (see PDF for code and example figure: https://submit.elifesciences.org/*eLife*_files/2022/05/19/00107349/00/107349_0_attach_6_474_convrt.pdf) that I hope will be useful to the authors.

2d. I recommend the authors show the full curve so that the readers get a more detailed sense of how averaging effects the power-law interpretation of the data.

3. Please check that this calculation and interpretation is correct.

4. Connection between brain and behavior:

4b. I recommend that the authors add distribution information to the plots in Figure~4B to give a sense of how points are spread through the [correlation with behavior]-by-[power law range] space. Potential plots might be a co-located histogram, or perhaps an uncertainty estimate as a function of correlation based on the number of points and variance. This would help show significance of the curves in a way that accounts for the uneven spread of datapoints.

4c. In a similar vein, what are the typical dynamic ranges for subsets correlated and uncorrelated with behavior? I recommend showing a number of these to see if those dynamic ranges are impacting the possible ranges in the [correlation with behavior]-by-[power law range] plots. Perhaps something like curve in each plot showing the minimum maximum value of the power law range per correlation range.

4d. In general I'm struggling with the interpretation of Figure~4b in the sense that there seems to be such variability between mice. How much do the authors feel that this is a difference in neural populations imaged, vs changes in imaging conditions (illumination, window clarity, optical alignment) or differences in mouse activity levels?

[Editors’ note: further revisions were suggested prior to acceptance, as described below.]

Thank you for resubmitting your work entitled "Scale-free behavioral dynamics directly linked with scale-free cortical dynamics" for further consideration by *eLife*. Your revised article has been evaluated by Timothy Behrens (Senior Editor) and a Reviewing Editor.

The manuscript has been improved, and the reviewers concurred that an eventual acceptance is likely, but there are some remaining issues that need to be addressed. Specifically, the reviewers think that it's important that you address the points raised by Reviewer #3 (see below) regarding the "Mechanisms vs. Statistics" questions.

*Reviewer #1 (Recommendations for the authors):*

The authors have responded effectively to previous reviews both in the updated writing (discussion and intro) as well as the models and results.

*Reviewer #2 (Recommendations for the authors):*

Overall the authors addressed my concerns adequately. The paper has important results and makes a valuable contribution.

*Reviewer #3 (Recommendations for the authors):*

I appreciate your time and effort to respond to my review points. I believe the majority of my points have been addressed. The main weakness I still see is the lack of discussion about the broader mechanisms beyond the basic structures described that could account for the observations (see below).

Mechanisms vs. Statistics

I would like to clarify my reason for bringing up this "statistical" viewpoint that I believe may have been lost in translation. The paper as it stands makes the following logical steps (in terms of the mechanistic model) 1) The data exhibits power law scaling under certain binning of neurons (effect E) and 2) One way to account for this effect E is to consider certain e-i models. This is reasonable, but the overall search for possible mechanisms seems to be a combination of intuition and trial and error. Different architectures were tried and either "passed" or "failed".

The purpose of bringing up the statistical properties needed was to hopefully raise a conversation around what core properties are needed to replicate this effect. As activity statistics and connectivity are intimately related in mechanistic models, such a characterization would point to how broad or narrow the family of mechanisms that are possible is: i.e. how significant is it that the given sampling of models in the papers points to certain configurations. The closest I can see is the discussion point here that I think fails to completely address this point:

"One possibility suggested by our model is that the scale-free dynamics we observe occur at the boundary between winner-less switching and single-winner locked-in dynamics (the red dashed line in Figures8E and F). Additional theoretical efforts are necessary to more fully explore how the traditional criticality hypothesis relates to the competitive criticality suggested by our model."

On the author's distinction between mechanisms and statistics, I do not believe the two are independent paths to choose between. Mechanisms have statistical signatures (so-called "statistical model") and data statistics inform which mechanisms are possible given data. At the end of the day, these are mathematical models that need to connect core concepts to data. My point for this particular paper, which I will try to say more clearly and succinctly now is that there are many possible mechanisms that could explain observed effect E. I was hoping in my prior review to spark a slightly longer discussion on what overall properties would such a family of mechanisms share. I fail to see how identifying core statistics that would pare down the possible mechanisms is at odds with looking for a mechanism that explains an effect.

---

## [Author Response]

Essential revisions:1. The reviewers would like to see a detailed and thoughtful reflection on the role that 3 Hz Ca imaging might play in the conclusions that the authors derive. While the dataset in question offers many neurons, this approach is, from other perspectives, impoverished – calcium intrinsically misses spikes, a 3 Hz sampling rate is two orders of magnitude slower than an action potential, and the recordings are relatively short for amassing substantial observations of low probability (large) avalanches. The potential concern is that some of this disconnect may reflect optophysiological constraints. One argument against this is that a truly scale free system should be observable at any temporal or spatial scale and still give rise to the same sets of power laws. This quickly falls apart when applied to biological systems which are neither infinite in time nor space. As a result, the severe mismatch between the spatial resolution (single cell) and the temporal resolution (3 Hz) of the dataset, combined with filtering intrinsic to calcium imaging, raises the possibility that the conclusions are influenced by the methods.

We quite agree with the reviewer that reconciling different scales of measurement is an important and interesting question. One clue comes from Stringer et al’s original paper (2019 Science). They analyzed time-resolved spike data (from Neuropixel recordings) alongside the Ca imaging data we analyzed here. They showed that if the ephys spike data was analyzed with coarse time resolution (300 ms time bins, analogous to the Ca imaging data), then the anticorrelated activity became apparent (50/50 positive/negative loadings of PC1). When analyzed at faster time scales, anticorrelations were not apparent (mostly positive loadings of PC1). This interesting point was shown in their Supplementary Fig 12.

This finding suggests that our findings about anticorrelated neural groups may be relevant only at coarse time scales. Moreover, this point suggests that avalanche statistics may differ when analyzed at very different time scales, because the cancelation of anticorrelated groups may not be an important factor at faster timescales.

In our revised manuscript, we explored this point further by analyzing spike data from Stringer et al 2019. We focused on the spikes recorded from one local population (one Neuropixel probe). We first took the spike times of ~300 neurons and convolved them with a fast rise/slow fall, like typical Ca transient. Then we downsampled to 3 Hz sample rate. Next, we deconvolved using the same methods as those used by Stringer et al (OASIS nonnegative deconvolution). And finally, we z-scored the resulting activity, as we did with the Ca imaging data. With this Ca-like signal in hand, we analyzed avalanches in four ways and compared the results. The four ways were: (1) the original time-resolved spikes (5 ms resolution), (2) the original spikes binned at 330 ms time res, (3) the full population of slow Ca-like signal, and (4) a correlated subset of neurons from the slow Ca-like signal. Based on the results of this new analysis (now in Figs S3 and S4), we found several interesting points that help reconcile potential differences between fast ephys and slow Ca signals:

1. In agreement with Sup Fig 12 from Stringer et al, anticorrelations are minimal in the fast, time-resolved spike data, but can be dominant in the slow, Ca-like signal.

2. Avalanche size distributions of spikes at fast timescales can exhibit a nice power law, consistent with previous results with exponents near -2 (e.g. Ma et al Neuron 2019, Fontenele et al PRL 2019). But, the same data at slow time scales exhibited poor power-laws when the entire population was considered together.

3. The slow time scale data could exhibit a better power law if subsets of neurons were considered, just like our main findings based on Ca imaging. This point was the same using coarse time-binned spike data and the slow Ca-like signals, which gives us some confidence that deconvolution does not miss too many spikes.

In our opinion, a more thorough understanding of how scale-free dynamics differs across timescales will require a whole other paper, but we think these new results in our Figs S3 and S4 provide some reassurance that our results can be reconciled with previous work on scale free neural activity at faster timescales.

2. Another reservation expressed by the referees has to do with the generality of the conclusions drawn from the mechanistic model. One of the connectivity motifs identified appears to be i+ to e- and i- to e+, where potentially i+/i- are SOM and VIP (or really any specific inhibitory type) cells. The specific connections to subsets of excitatory cells appear to be important (based on the solid lines in Figure 8). This seems surprising: is there any experimental support for excitatory cells to preferentially receive inhibition from either SOM or VIP, but not both?

There is indeed direct experimental support for the competitive relationship between SOM, VIP, and functionally distinct groups of excitatory neurons. This was shown in the paper by Josh Trachtenberg’s group: Garcia-Junco-Clemente et al 2017. An inhibitory pull-push circuit in frontal cortex. Nat Neurosci 20:389–392. However, we emphasize that we also showed (lower left motif in Fig 8G) that a simpler model with only one inhibitory group is sufficient to explain the anticorrelations and scale-free dynamics we observe. We opted to highlight the model with two inhibitory groups since it can also account for the Garcia-Junco-Clemente et al results.

In the section where we describe the model, we state, “We considered two inhibitory groups, instead of just one, to account for previous reports of anticorrelations between VIP and SOM inhibitory neurons in addition to anticorrelations between groups of excitatory neurons (Garcia-Junco-Clemente et al., 2017).”

More broadly, there was concern that the neat diagrams drawn here are misleading. The sample raster, showing what appears to be the full simulation, certainly captures the correlated/anti-correlated pattern of the 100 cells most correlated with a seed cell and 100 cells most anti-correlated with it, but it does not contain the 11,000 cells in between with zero to moderate levels of correlation.

We agree that our original model has several limitations and that one of the most obvious features lacking in our model is asynchronous neurons (The limitations are now discussed more openly in the last paragraph of the model subsection). In the data from the Garcia-Junco-Clemente et al paper above there are many asynchronous neurons as well. To ameliorate this limitation, we have now created a modified model that now accounts for asynchronous neurons together with the competing anticorrelated neurons (now shown and described in Fig 8 – figure supplement 2). We put this modified model in supplementary material and kept the simpler, original model in the main findings of our work, because the original model provides a simpler account of the features of the data we focused on in our work – i.e. anticorrelated scale-free fluctuations. The addition of the asynchronous population does not substantially change the behavior of the two anticorrelated groups in the original model.

We probably expect that the full covariance matrix has similar structure from any seed (see Meshulam et al. 2019, PRL, for an analysis of scaling of coarse-grained activity covariance), and this suggests multiple cross-over inhibition constraints, which seem like they could be hard to satisfy.

We agree that it remains an outstanding challenge to create a model that reproduces the full complexity of the covariance matrix. We feel that this challenge is beyond the scope of this paper, which is already arguably squeezing quite a lot into one manuscript (one reviewer already suggested removing figures!).

We added a paragraph at the end of the subsection about the model to emphasize this limitation of the model as well as other limitations. This new paragraph says:

While our model offers a simple explanation of anticorrelated scale-free dynamics, its simplicity comes with limitations. Perhaps the most obvious limitation of our model is that it does not include neurons with weak correlations to both e+ and e- (those neurons in the middle of the correlation spectra shown in Fig 7B). In Fig 8 - figure supplement 2, we show that our model can be modified in a simple way to include asynchronous neurons. Another limitation is that we assumed that all non-zero synaptic connections were equal in weight. We loosen this assumption allowing for variable weights in Fig 8 - figure supplement 2, without changing the basic features of anticorrelated scale-free fluctuations. Future work might improve our model further by accounting for neurons with intermediate correlations.

The motifs identified in Figure 8 likely exist, but I am left with many questions of what we learned about connectivity rules that would account for the full distribution of correlations. Would starting with an Erdos-Renyi network with slight over-representation of these motifs be sufficient? How important is the homogeneous connection weights from each pool assumption – would allowing connection weights with some dispersion change the results?

First, we emphasize that our specific goal with our model was to identify a possible mechanism for the anticorrelated scale-free fluctuations that played the key role in our analyses. We agree that this is not a complete account of all correlations, but this was not the goal of our work. Nonetheless, our new modified model in Fig 8 - figure supplement 2 now accounts for additional neurons with weak correlations. However, we think that future theoretical/modeling work will be required to better account for the intermediate correlations that are also present in the experimental data.

We confirmed that an Erdo-Renyi network of E and I neurons can produce scale-free dynamics, but cannot produce substantial anticorrelated dynamics (Fig 8G, top right motif). Additionally, the parameter space study we performed with our model in Fig 8 showed that if the interactions between the two excitatory groups exceed a certain tipping point density, then the model behavior switches to behavior expected from an Erdos-Renyi network (Fig 8F). Finally, we have now confirmed that some non-uniformity of synaptic weights does not change the main results (Fig 8 - figure supplement 2). In the model presented in Fig 8 - figure supplement 2, the value of each non-zero connection weight was drawn from a uniform distribution [0,0.01] or [-0.01,0] for excitatory and inhibitory connections, respectively. All of these facts are described in the model subsection of the paper results.

3. Putting 2) another way, it's unclear why the averaging is required in the first place. This operation projects the entire population down in an incredibly lossy way and removes much of the complexity of the population activity.

Our population averaging approach is motivated by theoretical predictions and previous work. According to established theoretical accounts of scale-free population events (i.e. non-equilibrium critical phenomena in neural systems) such population-summed event sizes should have power law statistics if the system is near a critical point. This approach has been used in many previous studies of scale-free neural activity (e.g. all of those cited in the introduction in relation to scale-free neuronal avalanches). One of the main results of our study is that the existing theories and models of critical dynamics in neural systems fail to account for small subsets of neurons with scale-free activity amid a larger population that does not conform to these statistics. We could not make this conclusion if we did not test the predictions of those existing theories and models.

Second, the authors state that it is highly curious that subsets of the population exhibit power laws while the entire population does not. While the discussion and hypothesizing about different e-i interactions is interesting, it is possible that there's a discussion to be had on a much more basic level of whether there are topology independent explanations, such as basic distributions of correlations between neurons that can explain the subnetwork averaging. Specifically, if the correlation to any given neuron falls off, e.g., with an exponential falloff (i.e., a Gaussian Process type covariance between neurons), it seems that similar effects should hold. This type of effect can be easily tested by generating null distributions existing code bases. This is an important point since local (broadly defined) correlations of neurons implying the observed subnetwork behavior means that many mechanisms that have local correlations but don't cluster in any meaningful way could also be responsible for the local averaging effect.

We appreciate the reviewer’s effort, trying out some code to generate a statistical model. We agree that we could create such a *statistical model* that describes the observed distribution of pairwise correlations among neurons. For instance, it would be trivial to directly measure the covariance matrix, mean activities, and autocorrelations of the experimental data, which would, of course, provide a very good statistical description of the data. It would also be simple to generate more approximate statistical descriptions of the data, using multivariate gaussians, similar to the code suggested by the reviewer. However, we emphasize, this would not meet the goal of our modeling effort, which is mechanistic, not statistical. The aim of our model was to identify a possible biophysical mechanism from which emerge certain observed statistical features of the data. We feel that a statistical model is not a suitable strategy to meet this aim. Nonetheless, we agree with the reviewer that clusters with sharp boundaries (like the distinction between e+ an e- in our model) are not necessary to reproduce the cancelation of anticorrelated neurons. In other words, we agree that sharp boundaries of the e+ and e- groups of our model are not crucial ingredients to match our observations.

4. In general, the discussion of "two networks" seems like it relies on the correlation plot of Figure~7B. The decay away from the peak correlation is sharp, but there does not seem to be significant clustering in the anti-correlation population, instead a very slow decay away from zero. The authors do not show evidence of clustering in the neurons, nor any biophysical reason why e and i neurons are present in the imaging data.

First a small reminder: As stated in the paper, the data here is only showing activity of excitatory neurons. Inhibitory neurons are certainly present in V1, but they are not recorded in this data set. Thus we interpret our e+ and e- groups as two subsets of anticorrelated excitatory neurons, like those we observed in the experimental data. We agree that our simplified model treats the anticorrelated subsets as if they are clustered, but this clustering is certainly not required for any of the data analyses of experimental data. We expect that our model could be improved to allow for a less sharp boundary between e+ and e- groups, but we leave that for future work, because it is not essential to most of the results in the paper. This limitation of the model is now stated clearly in the last paragraph of the model subsection.

The alternative explanation (as mentioned in (b)) is that the there is a more continuous set of correlations among the neurons with the same result. In fact one of the reviewers tested this themself using code to generate some data with the desired statistics, and the distribution of events seems to also describe this same observation. Obviously, the full test would need to use the same event identification code, and so it is quite important that the authors consider the much more generic explanation for the sub-network averaging effect. We recommend assessing the possibility that broader explanations, e.g., in the form of the distributions of correlations accounts for the observed phenomenon. Even with 10K neurons, there are many other forces at play influencing the observed network and while it is nice that e-i networks are one explanation, much less constraining explanations that are still biophysically feasible should be discussed and compared against. I have provided one possible approach (see PDF for code and example figure: https://submit.elifesciences.org/eLife_files/2022/05/19/00107349/00/107349_0_attach_6_474_convrt.pdf).

As discussed above, we respectfully disagree that a statistical model is an acceptable replacement for a mechanistic model, since we are seeking to understand possible biophysical mechanisms. A statistical model is agnostic about mechanisms. We have nothing against statistical models, but in this case, they would not serve our goals.

To emphasize our point about the inadequacy of a statistical model for our goals, consider the following argument. Imagine we directly computed the mean activities, covariance matrix, and autocorrelations of all 10000 neurons from the real data. Then, we would have in hand an excellent statistical model of the data. We could then create a surrogate data set by drawing random numbers from a multivariate gaussian with same statistical description (e.g. using code like that offered by reviewer 3). This would, by construction, result in the same numbers of correlated and anticorrelated surrogate neurons. But what would this tell us about the biophysical mechanisms that might underlie these observations? Nothing, in our opinion.

5. Another important aspect here is how single neurons behave. It was not clear if single neurons were stated to exhibit a power law. If they do, then that would help in that there are different limiting behaviors to the averaging that pass through the observed stated numbers. If not, then there is an additional oddity that one must average neurons at all to obtain a power law. We recommend the authors show the full curve so that the readers get a more detailed sense of how averaging effects the power-law interpretation of the data.

We understand that our approach may seem odd from the point of view of central-limit-theorem-type argument. However, as mentioned above (reply R3b) and in our paper, there is a well-established history of theory and corresponding experimental tests for power-law distributed population events in neural systems near criticality. The prediction from theory is that the population summed activity will have power-law distributed events or fluctuations. That is the prediction that motivates our approach. In these theories, it is certainly not necessary that individual neurons have power-law fluctuations on their own. In most previous theories, it is necessary to consider the collective activity of many neurons before the power-law statistics become apparent, because each individual neurons contributes only a small part to the emergent, collective fluctuations. This phenomenon does not require that each individual neuron have power-law fluctuations.

At the risk of being pedantic, we feel obliged to point out that one cannot understand the peculiar scale-free statistics that occur at criticality by considering the behavior of individual elements of the system; hence the notion that critical phenomena are “emergent”. This important fact is not trivial and is, for example, why there was a Nobel prize awarded in physics for developing theoretical understanding of critical phenomena.

6. There is something that seems off about the range of \β values inferred with the ranges of \tau and $\α$. With \tau in [0.9,1.1], then the denominator 1-\tau is in [-0.1, 0.1], which the authors state means that \β (found to be in [2,2.4]) is not near \β_{crackling} = (\α-1)/(1-\tau). It seems as this is the opposite, as the possible values of the \β_{crackling} is huge due to the denominator, and so \β is in the range of possible \β_{crackling} almost vacuously. Was this statement just poorly worded?

The point here is that theory of crackling noise predicts that the fit value of beta should be equal to (1-alpha)/(1-tau). In other words, a confirmation of the theory would have all the points on the unity line in the rightmost panels of Fig9D and 9E, not scattered by more than an order of magnitude around the unity line. (We now state this explicitly in the text where Fig 9 is discussed.) Broad scatter around the unity line means the theory prediction did not hold. This is well established in previous studies of scale-free brain dynamics and crackling noise theory (see for example Ma et al Neuron 2019, Shew et al Nature Physics 2015, Friedman et al PRL 2012). A clearer single example of the failure of the theory to predict beta is shown in Fig 5A,B, and C.

7) It is not clear if there is more to what the authors are trying to say with the specifics of the scale free fits for behavior. Apparently, these results are used to motivate the neural studies, but aside from that, the details of those ranges don't seem to come up again.

The reviewer is correct, the primary point in Fig 2 is that scale-free behavioral statistics often exist. Beyond this point about existence, reporting of the specific exponents and ranges is just standard practice for this kind of analysis; a natural question to ask after claiming that we find scale behavior is “what are the exponents and ranges”. We would be remiss not to report those numbers.

Given that the primary connection between neuronal and behavioral activity seems to be Figure 4. The distribution of points in these plots seem to be very lopsided, in that some plots have large ranges of few-to-no data points.

We agree that this whitespace in the figure panels is a somewhat awkward, but we chose to keep the horizontal axis the same for all panels of Fig 4B, because this shows that not all behaviors, and not all animals had the same range of behavioral correlations. We felt that hiding this was a bit misleading, so we kept the white space.

It would be very helpful to get a sense of the distribution of points that are a bit hard to see given the overlapping points and super-imposed lines. We recommend that the authors add distribution information to the plots in Figure 4B to give a sense of how points are spread through the [correlation with behavior]-by-[power law range] space. Potential plots might be a co-located histogram, or perhaps an uncertainty estimate as a function of correlation based on the number of points and variance. This would help show significance of the curves in a way that accounts for the uneven spread of datapoints.

We also agree that characterizing uncertainty in power law range as a function of correlation and providing distribution information are good ideas. If we have not misunderstood the reviewer’s suggestion, we think we have already done such characterization. The three lines in each panel of Fig 4B meet the goal of characterizing variability as a function of correlation. The middle line is median, the top and bottom lines span the quartile range, which is a good way to characterize variability around the median for non-normally distributed variability. We also provide information on how points are distributed by plotting the points with partially transparent markers. In this way, higher density of overlapping points creates darker regions in the clouds of points. We feel that this approach avoids hiding outlier points and accounts for distributions of overlapping points. Adding more information (actual distributions, e.g.) to these plots would be largely redundant and make them too cluttered. We added the following sentence to Fig 4 caption: “Points have partially transparent markers, thus darker areas reveal higher density of points.”

8. Neural activity correlated with some behavior variables can sometimes be the most active subset of neurons. This could potentially skew the maximum sizes of events and give behaviorally correlated subsets an unfair advantage in terms of the scale-free range. In a similar vain to 8), what are the typical dynamic ranges for subsets correlated and uncorrelated with behavior? We recommend showing a number of these to see if those dynamic ranges are impacting the possible ranges in the [correlation with behavior]-by-[power law range] plots. Perhaps something like curve in each plot showing the minimum maximum value of the power law range per correlation range.In general, the reviewers struggled with the interpretation of Figure 4b in the sense that there seems to be such variability between mice. How much do the authors feel that this is a difference in neural populations imaged, vs changes in imaging conditions (illumination, window clarity, optical alignment) or differences in mouse activity levels?

Here we follow a standard convention for Ca imaging data (e.g. original Stringer 2019 Science paper). Each neuron’s activity was z-scored before defining groups and events. This means that the “dynamic ranges” (ranges of activity amplitudes) of each single neuron were similar. This should “normalize” the maximum possible event size in a fair way. We note, however, that the primary reason this z-scoring is done is that the measurement signal-to-noise ratio can vary across neurons due to variable expression of fluorescent indicator molecules across neurons.

As mentioned in above, the scatter of points in Fig 4 panels already directly shows the min-to-max span of power law range for each correlation.

Reviewer #1 (Recommendations for the authors):This paper feels highly polished and thorough in its presentation. I truly enjoyed reading it and believe it will be of value to the community.

We truly appreciate the positive feedback. Thanks

A nuanced question: mouse #5 and mouse #6 consistently seems to break the rules implied by the rest of the dataset. The behavioral descriptors look normal in figure 2, but neural structure is notably different from the rest of the group in 3E, 4B, 6C, 7E, and S4 (that one is behavior). Is there anything meaningfully different about these recordings (n cells, mean event rate, anatomical location etc)?

We agree that 3 of the 9 recordings (mice 5-7) are somewhat “outlying” relative to the results from the other 6 recordings. We took a closer look at those recordings and found that at least one possible reason is that the animals were much more active than usual (mice 6 and 7 were 50 times more active the next most active mouse, in terms of median run speed) or much less activity than usual (mouse 5 was 10 time less active than the next least active mouse). We have pointed this out in the main text near the description of Figure 4 and showed it directly in Figure 4 —figure supplement 1.

Reviewer #2 (Recommendations for the authors):Comments, questions, and technical issues for the authors.1. The values of tau (size exponent) from neural avalanches are surprisingly low. For instance, in the Ma et al. (Hengen) 2019 Neuron paper, tau ranged from 1.5 to 1.9, sometimes more than 2, but never less than 1. I am wondering how the imaging preprocessing affects the estimate of tau. For instance, if you started with simulated spiking activity that has avalanches with size pdf that go as a power law with exponent tau, and then use a forward model to generate "calcium imaging," and then apply deconvolution, z-scoring, and low-pass filtering, and then measure the avalanches again: what is the new tau?

We appreciate the suggestion to do some forward modeling of Ca-like signals from spikes. This is the approach we have now taken for FiguresS3 and S4.

2. This may be an ignorant question (apologies). The power law range is quantified in decibels (dB) throughout the paper; do you actually mean decades?

The typical definition of dB is the ratio of two log quantities and then multiplied by a factor of 10 or 20 (depending on the context). In our usage, we did the ratio of two log quantities, but we never multiplied by 10 or 20. To avoid confusion, we have replaced “dB” with “decades” throughout the paper.

3. Related to the set-up of the model, was there a reason that there are no adaptation mechanisms in this network model, as there often are in mechanistic models for avalanche criticality (including past work by the authors of this paper, e.g. Shew's 2015 Nature Physics paper)? Also, there appears to have been an error with the reference manager, as this reference shows up twice in the reference list.

There was no reason for the omission of adaptation mechanisms other than to increase simplicity. We aimed to make the model as simple as possible and still explain the features of the data we were most concerned with (i.e. scale-free-ness and anticorrelations).

Thanks for catching the double reference. We fixed that now.

4. It would be helpful the authors could elaborate on predictions that their results make for future studies. Maybe this is rather technical, but can you tell us when you expect to find a power-law distribution as a function of how much of the population is sampled and for how long? What if you were analyzing Neuropixels data, where you lose the extensive spatial sampling (and the restriction to pyramidal cells only) but you gain 3 orders of magnitude in temporal resolution?

Please see above and Figure 3 —figure supplements 1-2.

5. On page 8, you ask "are all behavioral events equally correlated to their concurrent neural events, or are certain neural events from certain subsets of neurons more strongly related to behavioral events?" I don't understand the question. What does it mean for all behavioral events to be equally correlated to concurrent neural events? Aren't "concurrent neural events" in specific subsets of neurons?

We agree that the wording of this sentence was confusing, so we deleted it. We think the goals of the analysis in Figure 6 are clearer with this deletion.

Reviewer #3 (Recommendations for the authors):1. Limits of calcium imaging: My recommendation is to assess mathematically the potential impact of missing data on the range and power-law slope estimates, which are the primary values used throughout the paper.

We appreciate the concern here and agree that event size statistics could in principle be biased in some systematic way due to missed spikes due to deconvolution of Ca signals. To directly test this possibility, we performed a new analysis of spike data recorded with high time resolution electrophysiology. We began with forward-modeling process to create a low-time-resolution, Ca-like signal, using the same deconvolution algorithm (OASIS) that was used to generate the data we analyzed in our work here. In agreement with the reviewer’s concern, we found that spikes were sometimes missed, but the loss was not extreme and did not impact the neural event size statistics in a significant way compared to the ground truth we obtained directly from the original spike data (with no loss of spikes). This new work is now described in a new paragraph at the end of the subsection of results related to Fig 3 and in a new Fig 3 - figure supplement 1. The new paragraph says…

“Two concerns with the data analyzed here are that it was sampled at a slow time scale (3 Hz frame rate) and that the deconvolution methods used to obtain the data here from the raw GCAMP6s Ca imaging signals are likely to miss some activity (Huang et al., 2021). Since our analysis of neural events hinges on summing up activity across neurons, could it be that the missed activity creates systematic biases in our observed event size statistics? To address this question, we analyzed some time-resolved spike data (Neuropixel recording from Stringer et al 2019). Starting from the spike data, we created a slow signal, similar to that we analyzed here by convolving with a Ca-transient, down sampling, deconvolving, and z-scoring (Fig 3 - figure supplement 1). We compared neural event size distributions to “ground truth” based on the original spike data (with no loss of spikes) and found that the neural event size distributions were very similar, with the same exponent and same power-law range (Fig 3 - figure supplement 1). Thus, we conclude that our reported neural event size distributions are reliable.”

However, although loss of spikes did not impact the event size distributions much, the time-scale of measurement did matter. As discussed above and shown in Fig 3 - figure supplement 2, changing from 5 ms time resolution to 330 ms time resolution does change the exponent and the range of the power law. However, in the test data set we worked with, the existence of a power law was robust across time scales.

2. Correlations and power-laws in subsets.2a-c. My recommendation is to assess the possibility that broader explanations, e.g., in the form of the distributions of correlations accounts for the observed phenomenon. Even with 10K neurons, there are many other forces at play influencing the observed network and while it is nice that e-i networks are one explanation, much less constraining explanations that are still biophysically feasible should be discussed and compared against. I have provided one possible approach (see PDF for code and example figure: https://submit.elifesciences.org/eLife_files/2022/05/19/00107349/00/107349_0_attach_6_474_convrt.pdf) that I hope will be useful to the authors.

discussed above, we respectfully disagree that a statistical model is an acceptable replacement for a mechanistic model, since we are seeking to understand possible biophysical mechanisms. A statistical model is agnostic about mechanisms. We have nothing against statistical models, but in this case, they would not serve our goals.

To emphasize our point about the inadequacy of a statistical model for our goals, consider the following argument. Imagine we directly computed the mean activities, covariance matrix, and autocorrelations of all 10000 neurons from the real data. Then, we would have in hand an excellent statistical model of the data. We could then create a surrogate data set by drawing random numbers from a multivariate gaussian with same statistical description (e.g. using code like that offered by reviewer 3). This would, by construction, result in the same numbers of correlated and anticorrelated surrogate neurons. But what would this tell us about the biophysical mechanisms that might underlie these observations? Nothing, in our opinion.

2d. I recommend the authors show the full curve so that the readers get a more detailed sense of how averaging effects the power-law interpretation of the data.

It is not clear to us what is meant by the “full curve”.

3. Please check that this calculation and interpretation is correct.

The point here is that theory of crackling noise predicts that the fit value of beta should be equal to (1-alpha)/(1-tau). In other words, a confirmation of the theory would have all the points on the unity line in the rightmost panels of Fig9D and 9E, not scattered by more than an order of magnitude around the unity line. (We now state this explicitly in the text where Fig 9 is discussed.) Broad scatter around the unity line means the theory prediction did not hold. This is well established in previous studies of scale-free brain dynamics and crackling noise theory (see for example Ma et al Neuron 2019, Shew et al Nature Physics 2015, Friedman et al PRL 2012). A clearer single example of the failure of the theory to predict beta is shown in Fig 5A,B, and C.

4. Connection between brain and behavior:4b. I recommend that the authors add distribution information to the plots in Figure~4B to give a sense of how points are spread through the [correlation with behavior]-by-[power law range] space. Potential plots might be a co-located histogram, or perhaps an uncertainty estimate as a function of correlation based on the number of points and variance. This would help show significance of the curves in a way that accounts for the uneven spread of datapoints.

We also agree that characterizing uncertainty in power law range as a function of correlation and providing distribution information are good ideas. If we have not misunderstood the reviewer’s suggestion, we think we have already done such characterization. The three lines in each panel of Figure 4B meet the goal of characterizing variability as a function of correlation. The middle line is median, the top and bottom lines span the quartile range, which is a good way to characterize variability around the median for non-normally distributed variability. We also provide information on how points are distributed by plotting the points with partially transparent markers. In this way, higher density of overlapping points creates darker regions in the clouds of points. We feel that this approach avoids hiding outlier points and accounts for distributions of overlapping points. Adding more information (actual distributions, e.g.) to these plots would be largely redundant and make them too cluttered. We added the following sentence to Figure 4 caption:

“Points have partially transparent markers, thus darker areas reveal higher density of points.”

4c. In a similar vein, what are the typical dynamic ranges for subsets correlated and uncorrelated with behavior? I recommend showing a number of these to see if those dynamic ranges are impacting the possible ranges in the [correlation with behavior]-by-[power law range] plots. Perhaps something like curve in each plot showing the minimum maximum value of the power law range per correlation range.

Here we follow a standard convention for Ca imaging data (e.g. original Stringer 2019 Science paper). Each neuron’s activity was z-scored before defining groups and events. This means that the “dynamic ranges” (ranges of activity amplitudes) of each single neuron were similar. This should “normalize” the maximum possible event size in a fair way. We note, however, that the primary reason this z-scoring is done is that the measurement signal-to-noise ratio can vary across neurons due to variable expression of fluorescent indicator molecules across neurons.

As mentioned in R3j, the scatter of points in Figure 4 panels already directly shows the min-to-max span of power law range for each correlation.

4d. In general I'm struggling with the interpretation of Figure~4b in the sense that there seems to be such variability between mice. How much do the authors feel that this is a difference in neural populations imaged, vs changes in imaging conditions (illumination, window clarity, optical alignment) or differences in mouse activity levels?

We agree that 3 of the 9 recordings (mice 5-7) are somewhat “outlying” relative to the results from the other 6 recordings. We took a closer look at those recordings and found that at least one possible reason is that the animals were much more active than usual (mice 6 and 7 were 50 times more active the next most active mouse, in terms of median run speed) or much less activity than usual (mouse 5 was 10 time less active than the next least active mouse). We have pointed this out in the main text near the description of Fig 4 and showed it directly in Fig 4 – figure supplement 1.

[Editors’ note: further revisions were suggested prior to acceptance, as described below.]

Reviewer #3 (Recommendations for the authors):I appreciate your time and effort to respond to my review points. I believe the majority of my points have been addressed. The main weakness I still see is the lack of discussion about the broader mechanisms beyond the basic structures described that could account for the observations (see below).Mechanisms vs. StatisticsI would like to clarify my reason for bringing up this "statistical" viewpoint that I believe may have been lost in translation. The paper as it stands makes the following logical steps (in terms of the mechanistic model) (1) The data exhibits power law scaling under certain binning of neurons (effect E) and (2) One way to account for this effect E is to consider certain e-i models. This is reasonable, but the overall search for possible mechanisms seems to be a combination of intuition and trial and error. Different architectures were tried and either "passed" or "failed".The purpose of bringing up the statistical properties needed was to hopefully raise a conversation around what core properties are needed to replicate this effect. As activity statistics and connectivity are intimately related in mechanistic models, such a characterization would point to how broad or narrow the family of mechanisms that are possible is: i.e. how significant is it that the given sampling of models in the papers points to certain configurations. The closest I can see is the discussion point here that I think fails to completely address this point:"One possibility suggested by our model is that the scale-free dynamics we observe occur at the boundary between winner-less switching and single-winner locked-in dynamics (the red dashed line in Figures8E and F). Additional theoretical efforts are necessary to more fully explore how the traditional criticality hypothesis relates to the competitive criticality suggested by our model."On the author's distinction between mechanisms and statistics, I do not believe the two are independent paths to choose between. Mechanisms have statistical signatures (so-called "statistical model") and data statistics inform which mechanisms are possible given data. At the end of the day, these are mathematical models that need to connect core concepts to data. My point for this particular paper, which I will try to say more clearly and succinctly now is that there are many possible mechanisms that could explain observed effect E. I was hoping in my prior review to spark a slightly longer discussion on what overall properties would such a family of mechanisms share. I fail to see how identifying core statistics that would pare down the possible mechanisms is at odds with looking for a mechanism that explains an effect.

We appreciate the clarifications and agree that mechanisms and statistics are intimately related. Indeed, in this context, much of our main results (event size and duration distributions, correlation spectra, etc.) can be viewed as statistical properties. Admittedly, these are rather “high level” statistics. To expand on this, we have now quantified some more basic statistics. We have now added a supplementary figure with single neuron variance and pairwise covariance values, comparing between neural subsets that exhibited good power laws and those that did not (Figure 8 —figure supplement 3). We show in this new figure that neural subsets with large power-law range tended to be composed of neurons with relatively high variance and high pairwise covariance. Perhaps these statistical properties will be shared by the family of mechanistic models suggested by the reviewer. However, in our view, it is beyond the scope of our manuscript to fully identify the necessary and sufficient statistical properties a model must generate to agree with our main results.